# Red Sea Coral Reef Monitoring Site in Sudan after 39 Years Reveals Stagnant Reef Growth, Continuity and Change

Sarah Abdelhamid [1,2,*,†], Götz B. Reinicke [1,*,†], Rebecca Klaus [1], Johannes Höhn [3], Osama S. Saad [4] and Görres Grenzdörffer [2,*]

1 Deutsches Meeresmuseum, Ocean Museum Germany, 18439 Stralsund, Germany; rebecca.klaus@gmail.com
2 Faculty of Agriculture and Environmental Sciences, University of Rostock, 18059 Rostock, Germany
3 Coralaxy GmbH, 18182 Bentwisch, Germany
4 Department of Biological Oceanography, Red Sea University, Port-Sudan 33312, Sudan; shurhabil@hotmail.com
* Correspondence: sarah.abdelhamid@somabay.com (S.A.); goetz.reinicke@meeresmuseum.de (G.B.R.); goerres.grenzdoerffer@uni-rostock.de (G.G.)
† These authors contributed equally to this work.

**Abstract:** Coral reefs off the coast of the Republic of Sudan are still considered to be among the most pristine reefs in the central Red Sea. The complex coastal fringing reefs, offshore banks, and shoals of Dungonab Bay in the north and Sanganeb atoll situated further to the south, about 23 km off the Sudanese mainland coast, were inscribed on the UNESCO World Heritage List in 2016. Due to their remote location and limited access, monitoring of the status of the reefs has been sporadic. Here, we present the results of a repeated large area photomosaic survey (5 m × 5 m plots) on the Sanganeb atoll, first established and surveyed in 1980, and revisited in 1991 and most recently in 2019. The 2019 survey recovered and reinstated the four original monitoring plots. Evaluation of photographic and video records from one photomosaic plot on the seaward slope of the atoll revealed general continuity of the overall community structure and composition over 39 years. Individual colonies of *Echinopora gemmacea* and *Lobophyllia erythraea* were recorded in the exact same positions as in the 1980 and 1991 plots. The genera *Acropora* and *Pocillopora* remain dominant, although in altered proportions. Shifts in composition were detected at the species level (e.g., increase in *Pocillopora verrucosa*, *Stylophora pistillata*, *Acropora hemprichii*, *Dipsastraea pallida*, and *Echinopora gemmacea*, decrease in *Acropora cytherea* and *A. superba*), in addition to changes in the extent of uncolonized substrate (e.g., increase from 43.9% in 1980 to 52.2% in 2019), and other scleractinian, hydrozoan, and soft coral living cover. While the temporal resolution only includes three sampling events over 39 years (1980, 1991, 2019), this study presents one of the longest time series of benthic community surveys available for the entire Red Sea. A semi-quantitative estimate of vertical reef growth in the studied test plot indicates a reduction in net accretion rates of more than 80%, from 2.27 to 2.72 cm/yr between 1980 and 1991 to 0.28–0.42 cm/yr between 1991 and 2019. We carefully conclude that the changes observed in the coral community in the plot in 2019 (*Acropora–Pocillopora* shift, increase in *Montipora* and calcareous algae) are representative of impacts at the community level, including rising sea surface temperatures and recent bleaching events.

**Keywords:** coral community; long-term monitoring; net reef growth; Sanganeb atoll; *Acropora*; *Pocillopora*

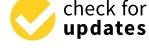



## 1. Introduction

The Red Sea hosts approximately 3.8% of the world's coral reefs [1]. The abiotic conditions prevailing in the central Red Sea region provide an optimal environment for coral growth and flourishing reef ecosystems [2]. The coral reefs of the Republic of Sudan are still considered to be among the most pristine and impressive reefs in the region. The reefs of Sanganeb support an impressive diversity of marine life, encompassing more than

251 species of fish (Red Sea, 1400 distinct fish taxa reported) and 124 cnidarian species (Red Sea, over 300 coral taxa) [3,4]. The only atoll within the Red Sea is situated at 19°45′ N and 37°26′ E, approximately 23 km east of the Sudanese mainland coast, and 28 km northeast of Port Sudan [5]. Sanganeb is an elongated oval-shaped reef, spanning approximately 6 km north–south and around 2 km east–west (Figure 1) [5]. Except for a navigable passage on the western side, the reef crest does not protrude above the water's surface, except during spring low tides, and there are only a few shallow channels across the reef.

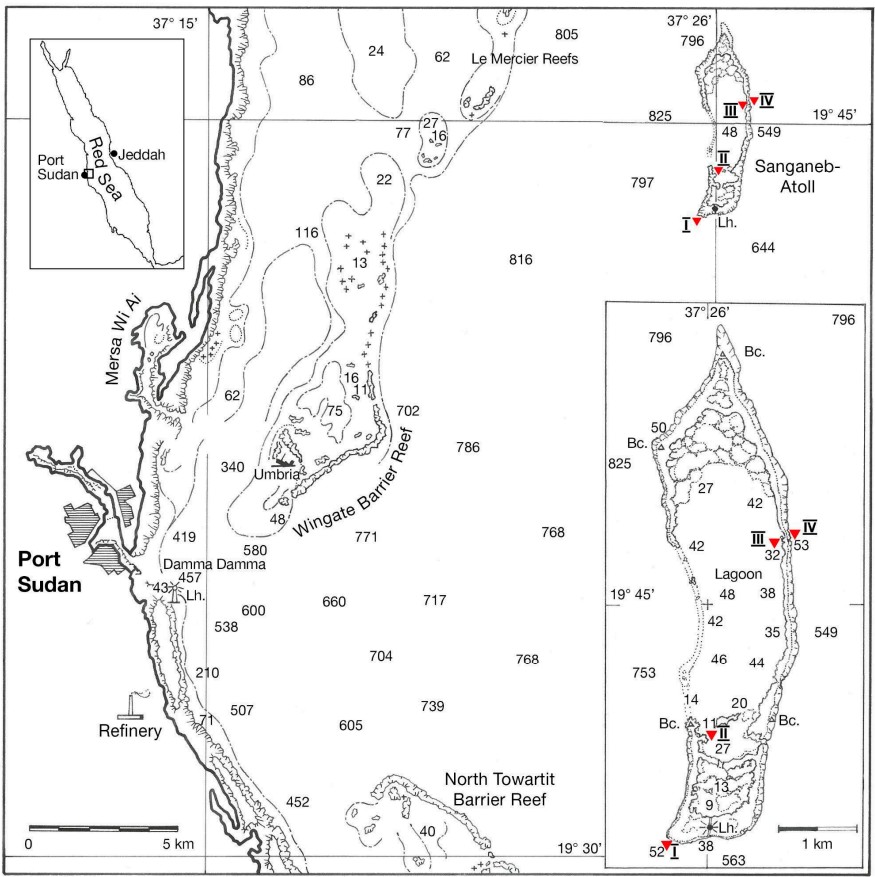

**Figure 1.** Map showing the location of Sanganeb atoll, about 20 km off the coast of Sudan, to the northeast of the city of Port Sudan (central Red Sea). Water depths are shown as small numbers. The inset map (bottom right) shows the position of the four test plots [TQs 1–4] around Sanganeb atoll, labelled I–IV. Lh = lighthouse, Bc = Beacon (after Mergner and Schuhmacher, 1985).

Designated as a UNESCO World Heritage site in 2016, the Sanganeb Marine National Park (SMNP) falls under the protection of the Wildlife Conservation General Administration (WCGA) [6], operating under the Sudanese Ministry of Interior [7]. Situated offshore, the atoll is not exposed to many of the land-based human influences that commonly affect inshore coral reefs. Factors such as contaminated sediment run-off or outfalls, and other types of physical disturbances arising from human activities within the coastal zone can be disregarded when considering the threats to the atoll's reef communities [5,8]. Moreover, the atoll is at low risk from coral predators (e.g., *Acanthaster* sp.) as well as other potential impacts, such as destructive fishing and aquaculture [7].

The SMNP hosts a variety of marine species, both commercially important and ecologically significant. The coral reef fish fauna in SMNP is highly diverse, with over 251 species identified to date, including the bumphead parrotfish (*Bolbometopon muricatum*) and at least nine species of grouper, with the spotted coral grouper being of particular commercial importance. Pelagic fish like tuna, barracuda, and sailfish are frequently encountered, as well as elasmobranchs such as manta rays, sharks, and marine mammals. The lagoon may

also be an important nursery and spawning ground. Notable invertebrates found around the SMNP include *Trochus dentatus* (accepted as *Tectus dentatus* [9]) and sea cucumbers, with *Tectus* populations appearing healthy and not over exploited. Research is ongoing to understand their natural distribution and abundance. The park also has pearl oysters, ornamental seashells, giant clams, sponges, nudibranch molluscs, and ascidians [4].

Consequently, the Sanganeb atoll coral reef is among the healthiest and most diverse reefs in the Red Sea [10]. Due to the remote location and limited access, monitoring of the status of Sudanese reefs has been sporadic. However, while minor impacts appear inconspicuous, detailed observations reveal certain anthropogenic influences [11].

The most substantial threat to the atoll's coral reefs stems from the effects of climate change, in particular elevated seawater temperatures that can trigger severe coral bleaching events [12]. Ongoing dynamic changes in coral communities are real and the state of most coral reefs is constantly altering, especially in areas subject to other anthropogenic impacts. Climate change is now the predominant threat. During the previous ten decades, the temperature of the tropical ocean has risen by approximately 1 °C. Projections indicate that this upward trend will persist, with an expected increase of 1–2 °C per century [2,13,14]. Between 1982 and 2006, the annual temperature average of the Red Sea has reportedly already increased by 0.74 °C [15,16], with an abrupt warming event in 1994, during which there was a 0.7 °C increase in mean annual sea surface temperatures (SST) [17]. Chaidez et al. (2017) calculated an overall trend toward increased $T_{max}$ values with $0.17 \pm 0.07$ °C decade$^{-1}$ in the Red Sea [18]. These large-scale alterations in the environmental conditions, including rising seawater temperatures due to climate change in particular, are inducing regional and long-term shifts in coral community health and composition [15].

Other studies that have investigated long-term changes in coral reefs of the Red Sea reflect the general dynamic status of coral communities. A reference site in the northern Gulf of Aqaba, named U7, was first inspected by Mergner and Schuhmacher in 1972 [19], and studied in detail in 1976, 1982, 1989, and 2005 over 29 years. Observations followed the fate of individual coral colonies and gross changes in the community composition. Drastic alterations (e.g., the almost entire disappearance of xeniid soft corals from the test plot) were recorded and related to ongoing coastal development activities in the immediate vicinity [20]. Other studies have examined the changes occurring within coral communities at the species level and quantified these over time [21–27]. In addition to temporal changes, other studies have also examined geographical shifts in species distribution and community composition within the Red Sea. A long-term study spanning 30 years revealed a decreasing trend in species richness from south to north in the Red Sea and concluded that communities exhibited increasing homogeneity across different latitudes [28]. This study presents the results for one of, if not the longest, time interval for a benthic community survey available for the entire Red Sea.

## 2. Materials and Methods

### *2.1. History and Methodology of Surveys*

The Sanganeb atoll was strategically selected as a suitable long-term monitoring site due to its remote location. The study commenced in 1980, when Hans Mergner and Helmut Schuhmacher, with their team, selected four test plots (then termed "test quadrats", TQ 1–4 in [5]) distributed along a transect across the Sanganeb atoll. The transect was oriented in a north–northeast (NNE) direction, started in the southeast corner of the atoll and crossed over the reef outline about 4 km further towards the NNE. The plots were selected with the intention of comparing coral communities with varying exposures to prevailing winds and current directions. All four test plots were located between 10 and 12 m water depth. This depth range was selected because it was hypothesised that it would best reflect the long-term effects of abiotic factors such as light, water currents, sedimentation, and biotic parameters [5]. Nylon ropes were used to demarcate a 5 × 5-m grid, subdivided into 1 m by 1 m sections. The size of the plots was selected based on a species–area curve, which

was calculated to determine the most appropriate minimum frame size for studying the coral community [5,24].

Helmut Schuhmacher and his team laid out and surveyed the test plots in the spring of 1980, and again in April 1991, when the plots were resurveyed to evaluate the community changes in all four TQs [5,24]. In 1980 and 1991, the surveys relied upon black and white (b/w) underwater photography, using Nikonos V cameras with flash in 1991, to photograph the 1 m$^2$ grids and corals during dozens of dives. Films were processed on-site in the Sanganeb lighthouse and b/w prints of the 1 m$^2$ grids were produced on waterproof paper for in situ use to map and identify coral colonies and species. The shapes of the coral colonies were traced by hand using transparent paper, and species-level identifications were determined in situ. For colonies that were challenging to identify underwater, samples were collected near the test area for later taxonomic identification based on their skeletons. Reference specimens of Scleractinia (including close-up colour slides) are today stored in the Ocean Museum Germany (OMG) Foundation collection.

Present Study

A survey of Sudanese reefs was carried out by the University of Vienna, in September–October 2019 onboard the MS "Don Questo". This survey included an attempt to recover and document the former TQs at the Sanganeb atoll. Götz B. Reinicke and the team successfully relocated the plots and temporarily re-established the nylon rope grid using old maps, photographs, and prominent or distinguishable coral colonies, as well as the remnants of original grid lines from the 1991 survey. While some sections of the older lines were overgrown or torn, many pieces were intact and in their original positions.

All four test plots were relocated using the published map locations. To support their relocation in future years, a hand-held geographical positioning system was used to record the geographical coordinates as follows: TQ1 19°43′368″ N, 37°26′236″ E; TQ2 19°44′017″ N, 37°26′571″ E; TQ3 19°45′324″ N, 37°27′123″ E, and TQ4 19°45′353″ N, 37°27′164″ E (Figure 1). The positions of TQs 3 and 4 were confirmed to be approximately 30 m south of the positions marked in the 1980 aerial photograph [5].

The 2019 survey re-evaluated all four monitoring sites as established in 1980 [5,24]. For the present study, only one site, plot TQ4, located on the eastern windward slope of the atoll was chosen for detailed analysis. TQ4 was selected as it was considered to be representative of the general conditions at the atoll and at the lower end of a fictional "gradient of chronic disturbance" among the four plots [24]. In TQ4, the prevailing structures formed by reef-building Cnidaria species appeared to be least impacted by local effects (e.g., lagoonal conditions and sedimentation). Detailed comparative analysis of the 1980 and 1991 TQ1–3 plots has been published [24], and the quantitative analysis of these plots in 2019 was therefore excluded from the current study due to the limited additional informational value.

Test plot TQ4 is located on the outer eastern slope of the atoll, and access to this site is influenced by sea conditions. This side of the reef is particularly exposed to the easterly swells from the open Red Sea, and wind-driven surface currents, creating strong swell and water movement. Wave heights at TQ4 were previously reported to reach mean values of at least 1.1 m during the survey period in February–March 1980 [5]. The complex current patterns in the central Red Sea result in a non-destructive, moderate to occasionally strong current regime, leaving almost no deposits of fine sediments in the plot at 10–12 m depth. Average monthly wave heights for the region were described below 1.0 m, mainly due to basin depth and overall limited distances of wind fetch (based on data 2008–2009) [29].

In 2019, after recovering and remarking the 5 m by 5 m plots, a survey team of three divers completed the photo-documentation survey in 2–3 dives per site using the protocols outlined by Sandin et al. (2019) [30]. The setup at each site involved the deployment of guidelines and calibration markers, the measurement of metadata for plot positioning, and camera calibration, as well as detailed documentation of coral colonies. One diver swam horizontally over the 5 m by 5 m plot, following the reef front and capturing photographs of 1 m wide sections centrally from 2 m above the seabed. Photographs were taken using a

handheld digital camera (Olympus TG5 camera) equipped with a timelapse serial function and a water level, in the full noonday sun. With the interval timer set to 5 s, the camera captured a row of 16–18 pictures for every 5 m long swim over the plot. This resulted in 8–11 overlapping rows and up to 180 photos per plot. The photo series was taken twice, capturing images in both 4:3 and 16:9 formats. Metadata about the plot measures and coordinates were recorded. Two other divers took additional photographs (including close ups) to document and support the taxonomic identification of individual coral colonies in the plot. Reference photographs of the coral colonies within the TQ4 are available upon request.

### 2.2. Ecological Processing of Large-Scale Image Documentation

The series of photographs captured in test plot TQ4 were merged to create an ortho-mosaic photo (Figure 2) using the software Metashape Professional v. 1.7.1 from Agisoft. Ground control was provided in two different ways. Four to six ground control points (GCPs) were placed along the edges and the corners of the plot. In addition, four pairs of markers with a fixed distance of 50 cm were placed in different parts of the sites. The distances between the GCPs in the corners of the plot were measured in situ with a tape and used to correct the orthomosaic. Due to the complex 3D topography within the plot, the tape measurements do not necessarily represent the exact distance. On average the tape measurements were 3–6 cm longer than in reality. The relative accuracy of the orthomosaics is within 3–5 mm, if only the four defined marker pairs are taken into account.

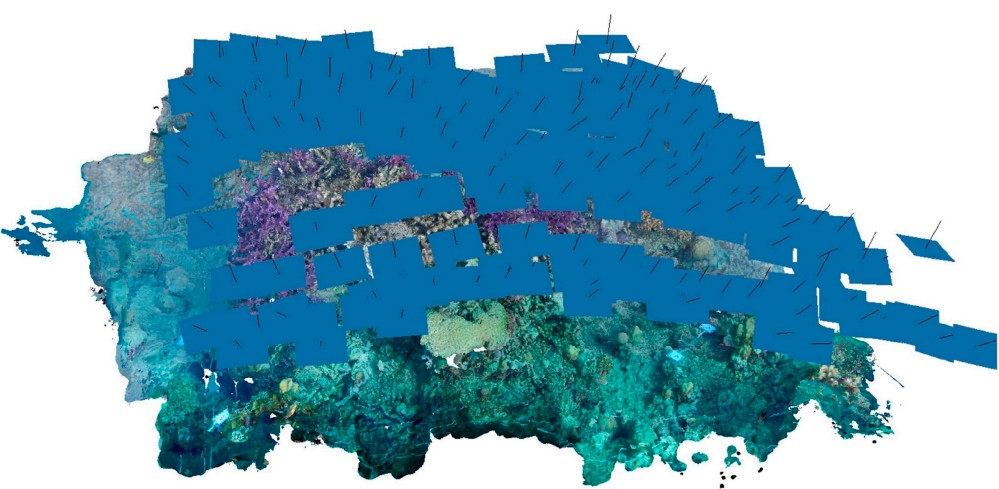

**Figure 2.** Matching and mosaicking of the series of individual images using Agisoft Metashape Professional v. 1.7.1 resulted in a digital orthophoto mosaic (see Figure 8).

The surface of the reefs is described with a precise 3D point cloud containing approximately 100 million points (as shown in Figure 2). Subsequently, a digital orthophoto mosaic was developed from these data, following the aforementioned protocol [30]. The orthomosaic was then used to examine benthic community composition in terms of coverage, as well as coral colony numbers, sizes, and species distribution (see Table S1)

### 2.3. Taxonomic Verification

The composition of the coral community in TQ4 was described at the species level where feasible (see Appendix A for the overall species list). In cases where the photographic resolution was blurred or sampling was incomplete, the determination was limited to the genus level. The identification of coral species documented during the 2019 survey primarily used resources [13,31] that were not available during the 1980 and 1991 surveys. Earlier resources were consulted for comparison [32,33]. To help ensure consistency with earlier observer IDs, the 1980 in situ colour slides and photographs were studied in detail, alongside the specimens in the 1980 reference collection at the OMG (Ocean Museum

Germany Foundation, Stralsund). The re-examination of the specimens and colour slides, previously determined by H. Schuhmacher, provided a broad foundation for re-identifying the specimens recorded in 1980 and verifying the consistency of the taxonomic work conducted in the present study. Three other scleractinian specialists were consulted in cases of doubt.

Despite this, certain colonies, particularly those belonging to the genera *Acropora*, *Porites*, and *Montipora* still posed a challenge in terms of species level identification because close-up photographs of each individual colony were not always available. Given this and the restriction on collecting reference samples from within or close to the test area, it was occasionally necessary to limit identification to the genus level. This was particularly relevant for the genus *Acropora*, which demonstrates a tendency for hybridization and is currently under revision within the Red Sea. Two reference specimens of the dominant *Acropora* sp. in the 1980 survey, then referred to as *A. superba*, were compared and re-identified (see section "Status of TQ4 in 2019"). Also, *Montipora* and *Porites* species, which are best differentiated by their minute corallite structure, were classified at the genus level for statistical analysis purposes.

### 2.4. Graphical Extraction of Community Data

The composition of the substrate in the orthomosaics was delineated using Adobe Photoshop software version 22.5.1 following the methodology outlined by Sandin et al. (2019) [30]. This involved outlining individual coral colonies and other substrate structures using a Wacom pen tablet and the pencil tool in Photoshop. A zoom level of 1130% and a pencil size of 3 pixels were employed during the outlining process. Each outlined substrate unit (polygon) was filled with a unique colour corresponding to its taxonomic classification. To ensure consistency and facilitate comparison with earlier substrate charts (1980, 1991), the same colour codes from previous studies were adopted and matched using the Photoshop eyedropper tool. Each substrate category was then assigned to a separate layer. To enhance species identification within the mosaic, additional original individual photos and close-up images of taxa were utilised to supplement missing information. Before exporting the data, thorough checks were conducted to identify and rectify any errors or inaccuracies, such as overpainting or incorrect colony placement within a layer. The finalised colour-coded layers were exported as individual PNG files from Photoshop and imported into the geographical information software ArcGIS Pro™ from ESRI (version 2.9.1). These PNG files were converted into polygons within ArcGIS Pro, with each polygon representing a specific taxon. The number and area of each polygon was calculated for each taxa.

The newly generated 2019 data were reformatted and integrated into the data tables prepared by Reinicke et al. (2003) [24] for analysis. This allowed for the comparison of key parameters, including the total cover, the relative proportion of each species, colony abundance, and colony sizes.

### 2.5. Net Reef Accretion Estimate

An estimate of net vertical reef accretion was determined using the overgrown remains of the grid lines from the previous surveys [24], p. 294. The vertical distances of line sections from each former grid laid out over the coral community framework in 1980, and again in 1991, were measured from the actual new grid levels in 1991 and 2019 (Figure 3). Estimates of the vertical elevation of the reef framework surface were then related to the elapsed time interval (in years) and the relative difference was calculated as a measure of vertical net reef growth (cm/yr):

$$reef\ accretion\ rate\ \left[\frac{cm}{yr}\right] = \frac{vertical\ distance\ between\ former\ and\ next\ gridlines\ [cm]}{time\ span\ between\ surveys\ [ys]}$$

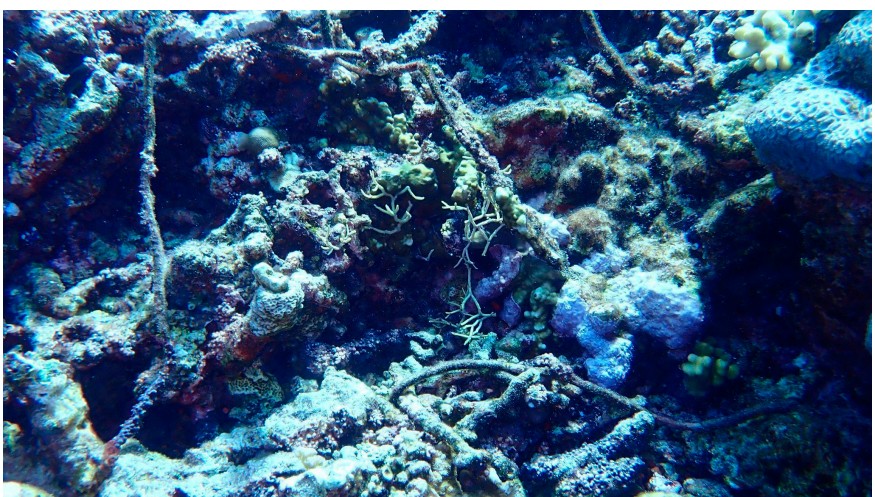

**Figure 3.** Remnants of the 1991 survey line grid within the TQ4 plot in 2019 (photo: G.B. Reinicke).

### 2.6. Data Analysis

Data analysis and visualisation were primarily conducted using Microsoft Excel version 16.70. Statistical tests and plots were performed using Excel Statistics (XLSTAT) with a significance level of $p < 0.05$. The Kolmogorov–Smirnov test was applied to all three time points to validate the normal distribution of the data. Based on the data distribution, Friedman's test, a non-parametric test for related samples, was utilised to determine if the populations from all three datasets originate from the same operational base. Additionally, a rank abundance plot was computed for visual representation.

Changes in the coral communities over time can be attributed to adaptation mechanisms in response to biotic and abiotic factors. These changes may occur on different time scales, influenced by factors such as seasonality, generational shifts, and specific events (e.g., bleaching). To understand the alterations in community structures within a habitat, it is meaningful to consider relative species turnover rates. In this study, species turnover was calculated for time scales of 11, 28, and 39 years, applying the formula of Schoener (1983) [34] to evaluate rates of change at the species level.

$$T_{rel} = \frac{(I_{abs} + E_{abs})100}{t\,(S1 + S2)}$$

where $I_{abs}$ = number of appearing species, $E_{abs}$ = number of disappearing species, $S1$ = total number of species in the start years of the periods (1980, 1980, 1991), $S2$ = total number of tested species at the end of the period (1991, 2019, 2019).

Estimates of dynamic changes to assess fluctuations in colony numbers and species coverage, as employed by Reinicke et al. (2003) [24] for the 1980 and 1991 sampling events, with an interval of 11 years, were again evaluated in this study. The prolonged 28-year interval between the 1991 and the 2019 sampling events appeared too long for robust conclusions about the fate of individual colonies.

Where the photographic resolution was blurred or samples were incomplete, the determination was limited to the genus level. Therefore, those taxonomic groups of Fungiidae, *Ovabunda*, *Porites*, and *Montipora* were generally counted as "1 taxon" in this calculation for all time intervals.

### 2.7. Multivariate Analysis

Multivariate analyses were performed using PRIMER 7.0 (Plymouth Marine Laboratory) to identify trends in the coral cover and coral colony data [35]. The data were initially explored using the 'Matrix display' routine, which produces a shade plot that is in effect an 'image' of the data matrix, with species abundances shown by depth of shading. The

shade plots help with the choice of pre-treatments (e.g., transformation), but also help with data interpretation.

Other multivariate analyses included the production of a Bray–Curtis similarity matrix based on both the raw data, standardised, and transformed data. Cluster dendrograms were produced and SIMPROF routines (using 1000 permutations, 95% significance level), were conducted in conjunction with these analyses.

In the results presented below, the results of the SIMPROF analyses are overlaid onto the dendrograms as red lines joining sites/samples that were statistically indistinguishable from one another. As a consequence of the permutative nature of the SIMPROF routine, only groups of three or more sites/samples may be considered as a true cluster, and groups of two sites/samples may be considered closely associated pairs. Individual sites/samples are considered outliers.

The data were also analysed with a SIMPER analysis to determine whether variability in the abundance of specific taxa was particularly important in determining the statistical distinctions made between clusters in the full dataset. The results of the SIMPER analyses are presented below.

## 3. Results

The methodologies employed in this 2019 study broadly followed those used during the earlier surveys in 1980 and 1991 [5,24] to maintain consistency with the long-term data sets and enable the tracking of changes over time. The ability to directly compare community composition at the species level was in part complicated by progress in coral taxonomy and systematics, noting that since the initial survey in 1980, a total of 124 cnidarian species have been identified within the SMNP [5,24]. Nevertheless, technological advancements in navigation, scuba diving, digital photography and videography, as well as available computing tools enhanced the quality and processing of data collected in 2019; hence, the comparative assessment across the surveys conducted in 1980, 1991, and 2019.

### 3.1. General Observations of Test Plots

The 2019 survey included the qualitative re-evaluation of the four monitoring plots originally established in 1980 (Mergner and Schuhmacher, 1985, [5]). Visual re-examination of TQ1 (Figure 4), TQ2 (Figure 5), and TQ3 (Figure 6) after 28 years confirmed the predominance of local impacts on the reef community structures within these plots; obvious sedimentation from downwind transport of sediments and coral debris from the reef flat above in TQ1 (Figure 4), impressive morphological alterations and elevation of the three-dimensional structure due to the extent of growth of the still predominant *Lobophyllia hemprichii* colonies in the upper half sector of TQ2 (Figure 5), and partial burying by coral rubble avalanching from the downwind lagoon inner slope of the atoll, covering about 15% of the bottom sector in TQ3 (see Figure 6, compare [24]). All colour-coded maps of the 1980 and 1991 surveys were plotted and described by Reinicke et al. (2003) [24], and reproduced for TQ4 at the end of the paper.

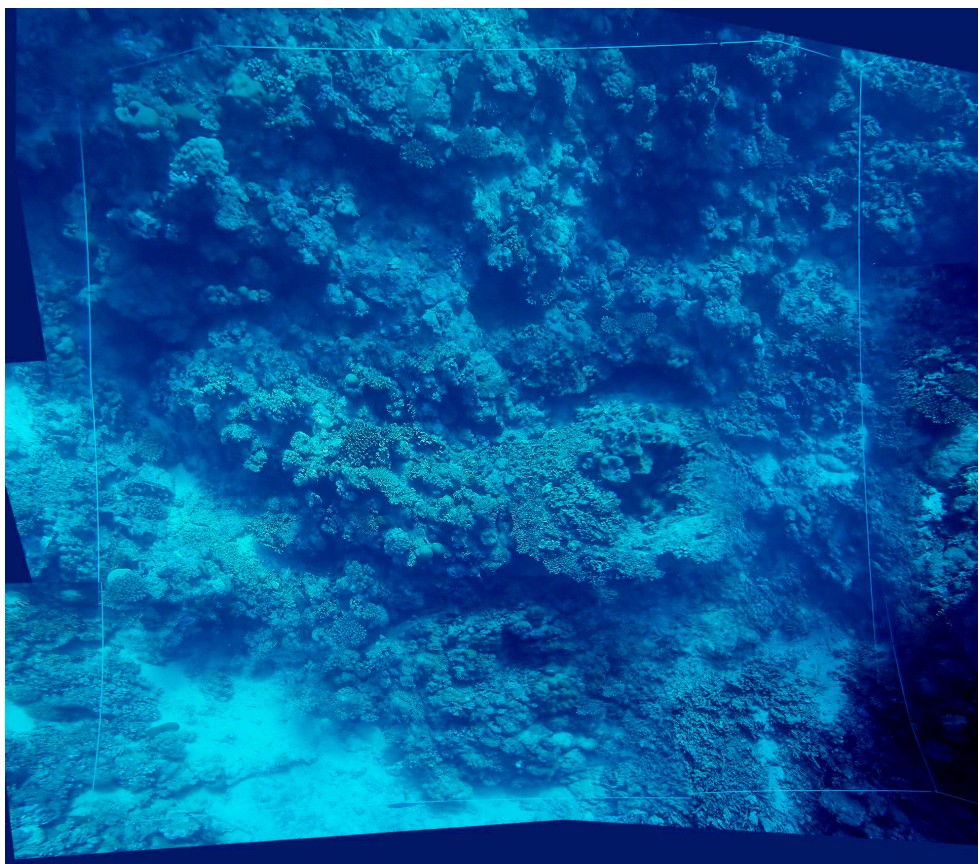

**Figure 4.** View of TQ1 on 3 October 2019. The site morphology appears unaltered compared to 1980, with sand and gravel loads drifting downslope on the bottom left and right sectors. Non-living substrata, including dead coral skeletons, dominate, while xeniid and other soft corals colonise current exposed positions, protected from sedimentation risk. Smaller, younger scleractinian colonies (*Acropora*, *Pocillopora*, *Porites* spp.) are common, reflecting frequent perturbation (line square 5 × 5 m, photo credit: G.B. Reinicke).

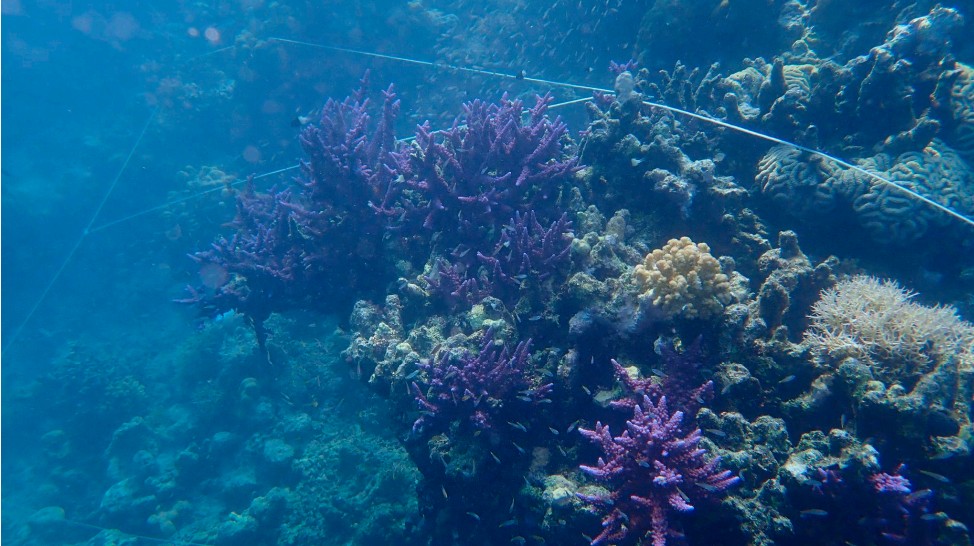

**Figure 5.** View of the upper sector of TQ2 on 30 September 2019. The original morphology of the sloping plot is characterised by the build-up of *Lobophyllia corymbosa* framework of about 1.5–2 m height (bottom left). Colonies of *Acropora hemprichii* occupy similar positions as in 1980 and 1991 (photo credit: G.B. Reinicke).

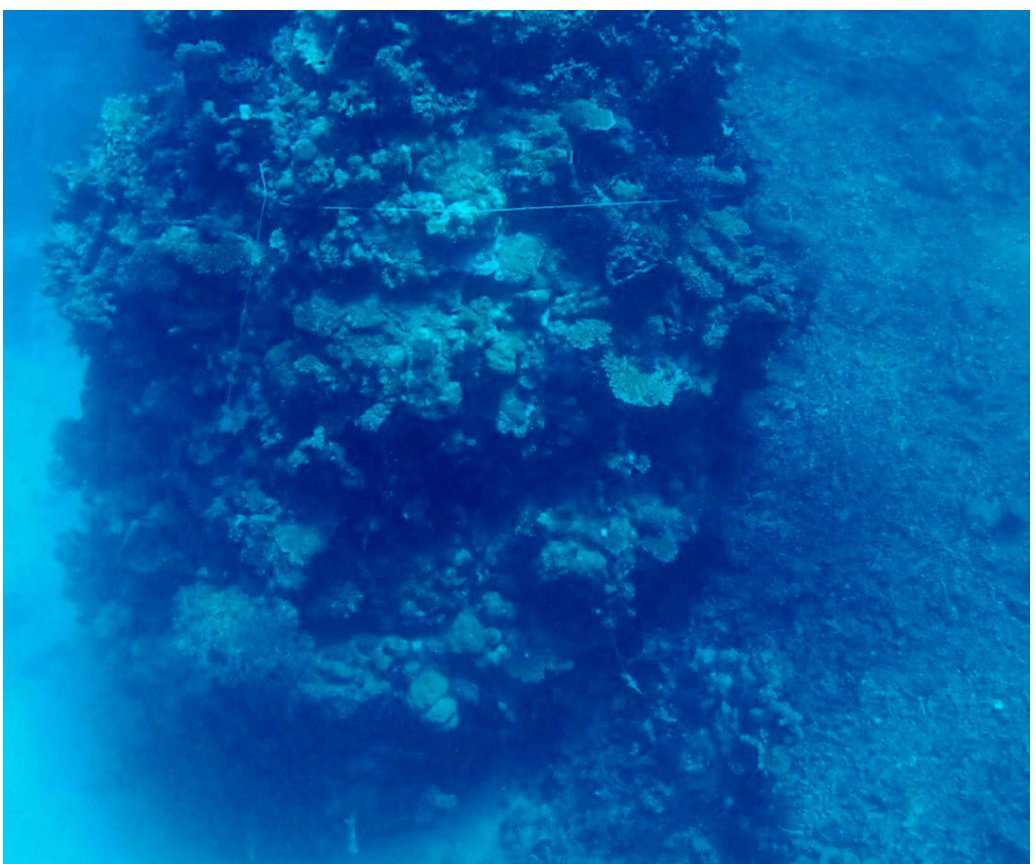

**Figure 6.** View of TQ3 on 3 October 2019. The morphology of the solitary substrate pillar has retained its shape and dimensions, the community is characterised by a lush and diverse cnidarian coverage; in the bottom right sector of the TQ, about 15% of the plot area was covered by coral rubble sliding down the atoll's lagoon side slope (video and photo: J. Höhn).

### 3.2. Status of the Test Plot TQ4 in 2019

#### 3.2.1. Topographic Description of the Surveyed Area

The position and orientation of a reef area, along with the prevailing water movement, play crucial roles in its settlement structure [5]. The reef flat and reef edge, which are exposed to wave action, are key zones for sediment creation, while the most protected area of the forereef is dominated by sediment accumulation. The TQ4 plot is located towards the east–northeast of the atoll, which is exposed to the average main direction of water movement, and access is dependent on relatively calm sea conditions. TQ4 spans a depth range of 8 to 11.5 m, with an average depth of 9.75 m. Below 13 m depth, the reef drops off to 52 m, maintaining a horizontal offset of only 2.5 m (see Figure 7). Despite underwater visibility exceeding 35 m, the lower section of the slope, composed of coarse sand, coral rubble, and mussel shingle, becomes obscured. This expansive scarp comprises three prominent overhangs; one between 16 and 23 m with a 105° inclination, another from 24 to 29 m with a 95° slope, and a third spanning 43 to 52 m with a 100° incline. The intervening sections exhibit an 85° slope and feature occasional coral boulders and sharp ledges. Notably, at a depth of 23 to 24 m, there is a striking 2 m-wide step, possibly indicative of a surf terrace established by lower sea levels during glacial periods. Below 52 m, the sediment accumulation displays limited signs of live benthic growth [5].

Based on the photo mosaic in Figure 8, Figure 9 maps the distribution of benthic structures and cnidarians colonising the substrate within TQ4 in 2019. Figure 20 shows the equivalent maps for all survey years (1980, 1991, and 2019).

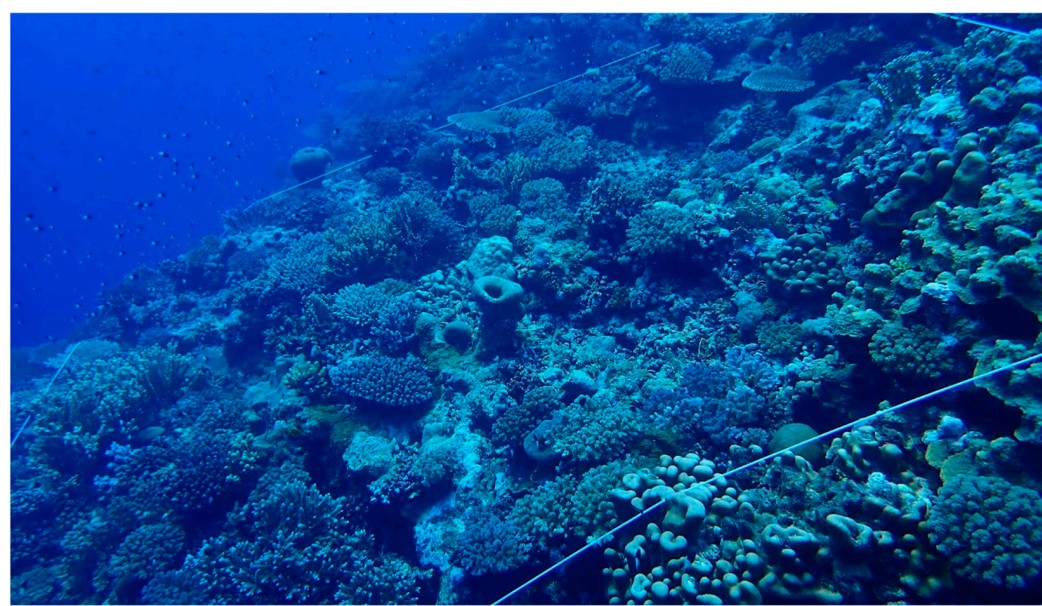

**Figure 7.** View of the TQ4 (from the north) on 1 October 2019. The substrate cover at the windward slope between 8 and 12 m depth shows a good representation of the upper slope habitats. Structural variability of the framework build-up is caused by underlying morphological anomalies like a deep trench (in the background) with gravel sloping about 2–3 m wide from a fracture in the reef crest above (photo credit: G.B. Reinicke).

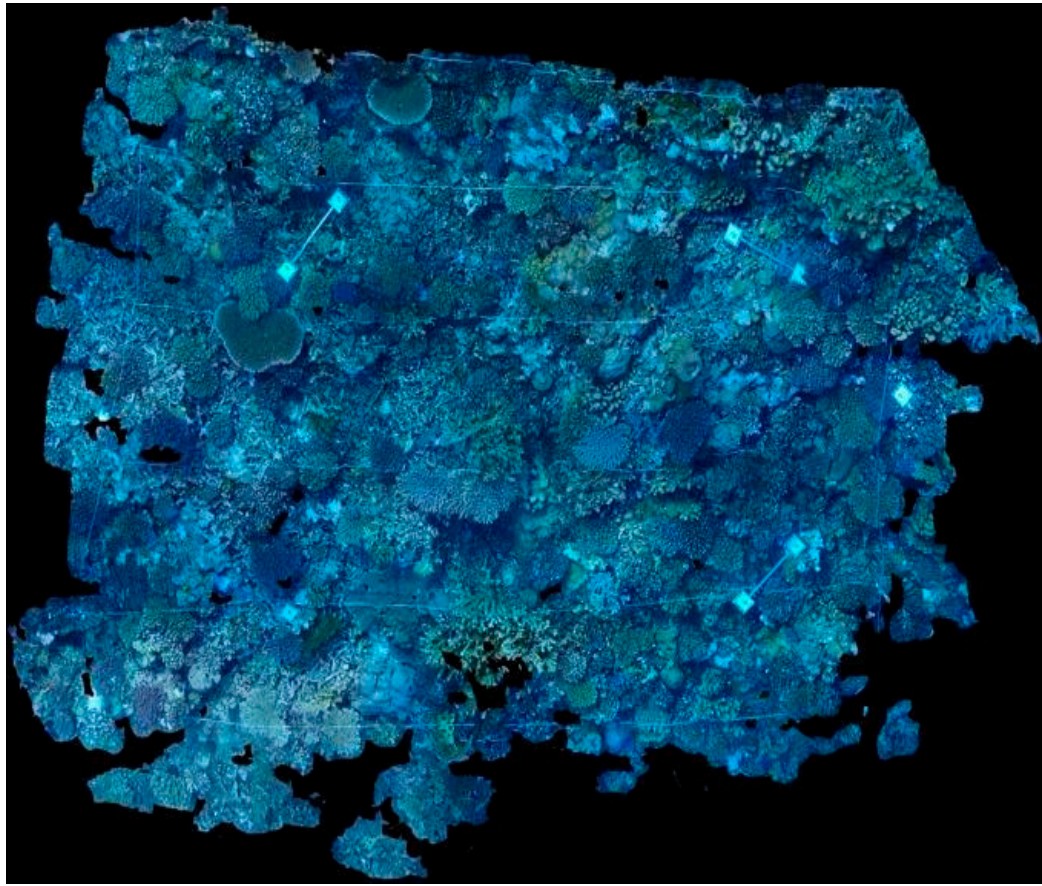

**Figure 8.** Mosaic orthophoto of test plot TQ4, Sanganeb atoll, all photos taken on 1 October 2019 (photo credit: G.B. Reinicke).

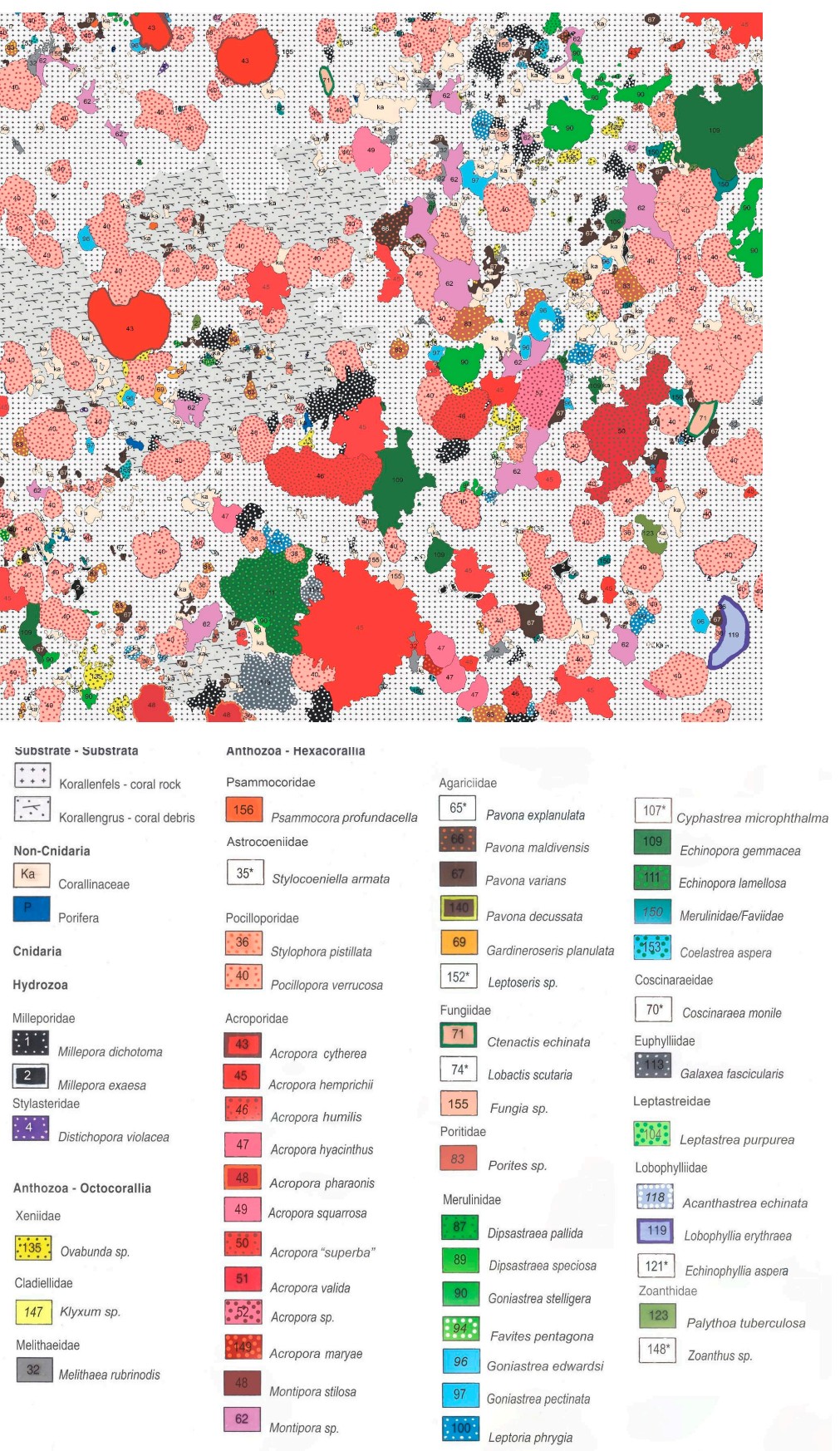

**Figure 9.** Colour-coded 5 × 5 m map of the test area TQ4 (NE–outer reef of the Sanganeb atoll) in the 2019 survey; Scale: c. 1:40. Legend of colour-coded map of the TQ4 plot in 2019 shows the

Hexacorallia: *Acropora* (red); Pocilloporidae (pink-red); *Psammocora* (orange); Fungiidae (salmon); *Montipora* (purple); Merulinidae (green to blue); Agariciidae (dark-brown); Oculinidae (grey). Hydrozoa: Milleporidae (black); Alcyonacea Alcyoniidae (yellow); Porifera (P), and Calcareous algae (ka). Taxa marked by an asterisk* were identified on close-up photographs but were too small to figure in the map.

Overall, in 2019, the percentage of the category "substrate" (including different components such as coral rock, dead coral colonies, and rubble, e.g., broken coral branches and coral debris) was lower in 1980 and 1991, and the share of living coverage was 53%.

The dominance of reef-building Cnidaria is visually evident in both Figures 9 and 20. Certain hydrocorals and stony corals stand out as leading species, re-affirming the fast-growing, reef-building scleractinians and hydrocorals as key elements in the community. The plot displays a dense and evenly distributed cover of *Pocillopora verrucosa* colonies (Figures 7–9). Indeed, the majority of the scleractinians present have a branching growth form. Closer inspection reveals an irregularity in the arrangement of the animate substrate. There is a change in the angle of the slope between the upper flatter part (25°) and the lower steeper part (40°) which occurs diagonally, spanning from the lower left to the upper right of the plot. This change in the angle of slope separates two differently illuminated areas, which is also reflected in the species present, with numerous *Millepora dichotoma* fans (black with white dots in Figure 9), *Echinopora* (dark green and dark green with paler green dots), and *Goniastrea* (blue) colonies growing along the junction between these areas. In addition, distinctive individual colonies of *Echinopora gemmacea* (#109, green, in the top right corner of Figure 9, shown in Figure 10A), and *Lobophyllia erythrea* (#119, purple, located in the bottom right corner of Figure 9, colony shown in Figure 10B) have successfully persisted throughout the entire 39-year investigation period.

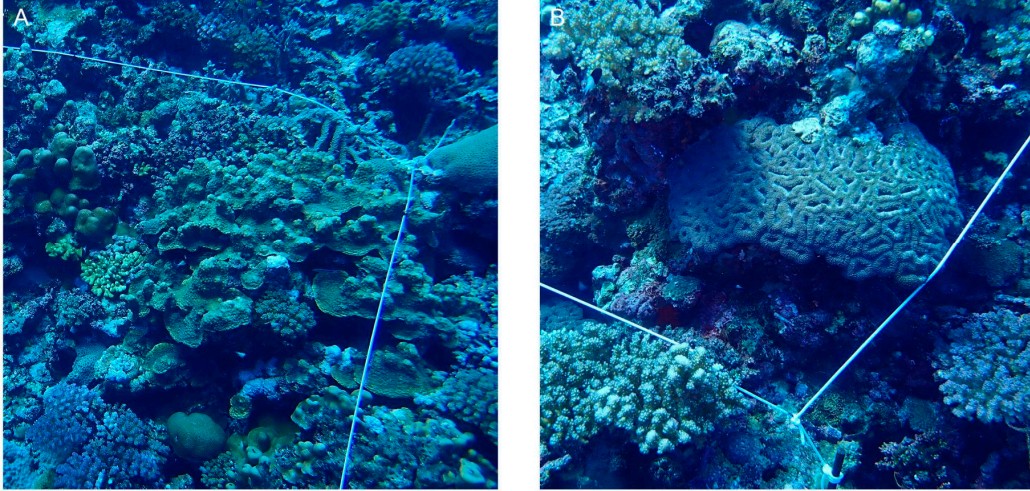

**Figure 10.** Colonies of *Echinopora gemmacea* (**A**, top right) and *Lobophyllia erythraea* (**B**, bottom right corners of TQ4 plot) have persisted in the same positions within TQ4 since 1980 (photo credit: G.B. Reinicke, 1 October 2019).

In 2019, *Pocillopora verrucosa* (38.7%) and *Acropora hemprichii* (10.4%) dominated TQ4, accounting for 49% of the total area colonised by cnidarians and represented by 86 colonies despite the presence of a total of 47 species in the test plot (with *Acropora "superba"* and *Pocillopora verrucosa* already being predominant species in 1980 and 1991). At least nine other *Acropora* species contributed an additional 12.9% of the cnidarian cover, represented by 40 colonies. *Pocillopora* spp. and *Acropora* spp. together account for about 62% of living coverage.

3.2.2. Net Reef Accretion during Survey Intervals

The net vertical accretion of the reef framework at TQ4 was assessed by examining the survey grid line levels within the plot for each survey interval (as depicted in Figure 11). In 1991, the recovered 1980 gridlines (red line in Figure 11) were 25–30 cm below the new 1991 grid line, which equates to a vertical accretion rate of between 2.27 to 2.72 cm/yr over the span of 11 years. In 2019, the recovered 1991 grid lines (green line in Figure 11) were 8–12 cm below the new 2019 grid lines on the current community framework (turquoise line in Figure 11), which equates to a vertical accretion rate of 0.28–0.42 cm/yr between 1991 and 2019 and the extended period of 28 years (Figure 11). These results indicate a near ten-fold reduction in vertical reef accretion.

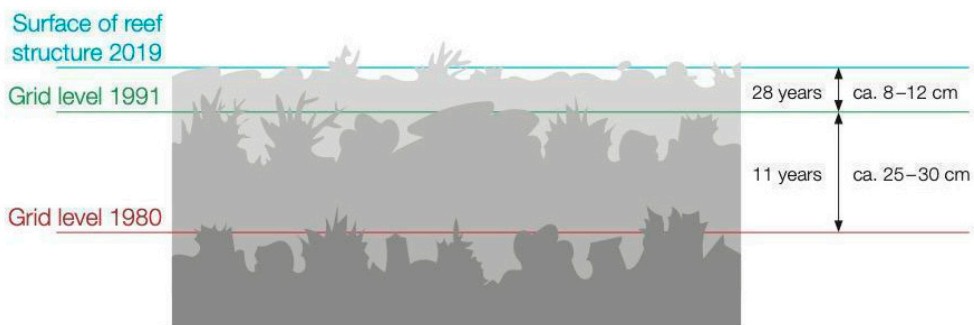

**Figure 11.** Schematic diagram to illustrate estimated vertical reef growth based on the surface levels during the three surveys (vertical axis to scale). Substrate overgrowth is shaded for 1980 (dark grey), 1991 (mid-grey), and 2019 (light grey). The first net grid level of 1980, recovered in 1991, is shown as a red line. The green line represents the net grid level of 1991 and the turquoise line represents the overgrowth of 8–12 cm between 1991–2019.

*3.3. Quantitative Evaluation of Coral Biocoenosis in 2019*

The data provide insights into the status and composition of the benthic community in TQ4 in 2019 and provide the foundation for a long-term evaluation of the warm water coral reef community dynamics spanning an interval of 39 years, from 1980 to 2019.

In 2019, a total of 47 cnidarian species belonging to 31 genera were identified (see Appendix A), with a total of 532 individual colonies recorded. These included three species from two genera of hydrozoans, two species from two genera of Octocorallia, and 42 species from 27 genera of Hexacorallia.

The data for the three surveys are categorised and presented in Table 1 and Figure 12A,B. Table 1 Part A presents the breakdown of benthic cover. Part B of Table 1 presents the proportions of Scleractinia, Hydroidea (including *Millepora dichotoma*, *Millepora exaesa*, *Millepora platyphylla*, *Distichopora violacea*), Alcyonaria (*Klyxum* sp., *Ovabunda* sp.), and other Cnidaria (*Melithea rubrinodis*, *Palythoa tuberculosa*). Part C provides information on the settlement by calcareous algae and Porifera, while Part D pertains to the proportional distribution of cnidarian growth forms, encompassing both hard and soft corals.

**Table 1.** Comparison of benthic cover data within test plot TQ4 on the Sanganeb atoll (Sudan) in 1980, 1991, and 2019, including earlier data from Reinicke et al. (2003) [24]. Minor deviations from summing up to 100% originate as inaccuracies due to rounding errors.

| Year of survey | 1980 | 1991 | 2019 |
|---|---|---|---|
| *A: Bottom coverage* | | | |
| Substrate | 43.9 | 55.5 | 52.2 |
| Cnidarian cover | 52.8 | 42.5 | 44.5 |
| Non-cnidarian cover | 2.5 | 2 | 3.3 |
| Sum*: total area cover | 99.2 | 100 | 100 |

**Table 1.** *Cont.*

| Year of survey | 1980 | 1991 | 2019 |
|---|---|---|---|
| *B: Coverage of Cnidarian Taxa (%)* | | | |
| Scleractinia | 45.2 | 32.4 | 41.7 |
| Hydroidea | 5.2 | 5.9 | 2.3 |
| Alcyonaria | 1 | 2.7 | 0.1 |
| Other Cnidaria | 1.4 | 1.5 | 0.6 |
| *C: Non-cnidarian cover (%)* | | | |
| Coralline algae | 1.9 | 1.1 | 3.2 |
| ## of patches | 53 | 51 | 286 |
| Porifera | 0.5 | 0.9 | 0.1 |
| ## of individual units | 149 | 70 | 13 |
| *D: Cnidarian growth forms, total hard and soft corals (%)* | | | |
| Branching corals | 38.3 | 31.7 | 28.4 |
| Massive corals | 6.3 | 3 | 4.9 |
| Encrusting corals | 5.6 | 3.2 | 7.3 |
| Total hard corals | 50.2 | 38.2 | 40.6 |
| Soft corals | 2.4 | 4.2 | 0.7 |

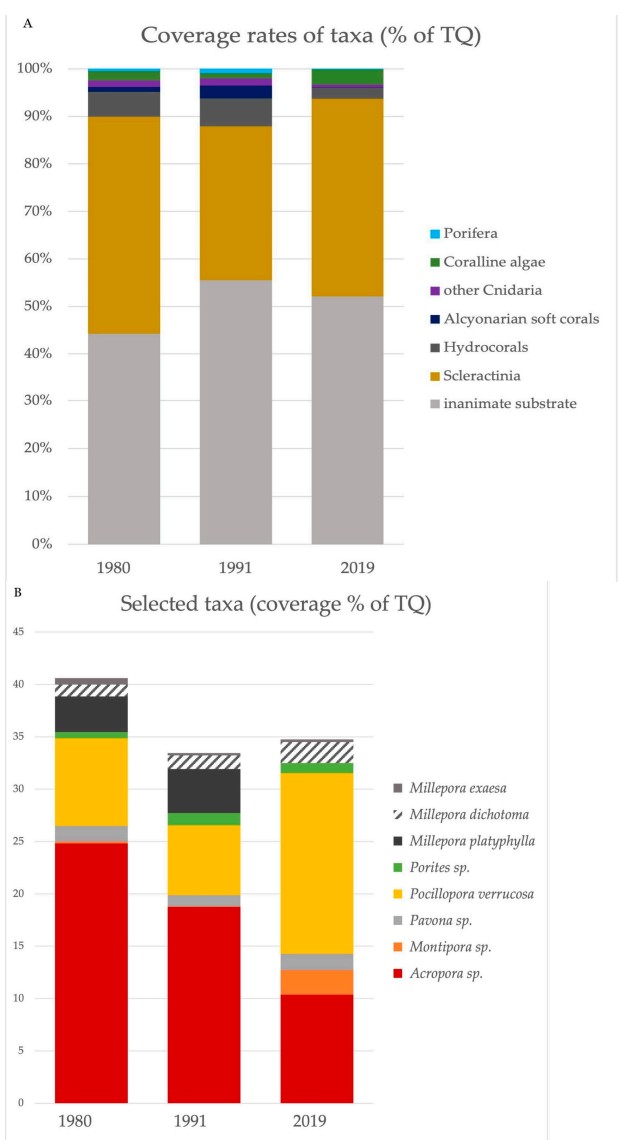

**Figure 12. (A)** Percent cover of main benthic categories including substrata, non-cnidarian, and cnidarian in the test plot TQ4 in 1980, 1991, and 2019 on the Sanganeb atoll. **(B)** Changes in the percent cover of selected cnidarian taxa in the test plot TQ4 in 1980, 1991, and 2019, respectively.

### 3.3.1. Changes in the Bottom Cover

The test plot's living (animate) and non-living (inanimate) substrate for the three surveys are categorised and presented in Table 1 part A and Figure 12A. Over the investigated period, the total live cover decreased from 56% to 45%, and 48% in 1980, 1991, and 2019, respectively. Cnidarian taxa dominated the live benthic coverage in all years, accounting for 44.5% in 2019, while non-cnidarian taxa comprised 3.3% in 2019.

The category "substrate" includes different non-living components such as coral rock, dead coral colonies, and rubble (e.g., broken coral branches and coral debris). Fine sediments (e.g., sand) were not observed. The total non-living cover increased from 43.9% in 1980 to 55.5% and 52.2% in 1991 and 2019, respectively.

In the map (Figure 9), non-living components are represented by a white pattern with distinct symbols, including small crosses for coral rock and dead corals, as well as dots and branches symbolising loose gravel. Among the inanimate substrate, coral rock was dominant (42.8%), while coral debris covered 9.4% of the test area, without sandy patches.

### 3.3.2. Changes in the Coverage of Cnidaria and Non-Cnidaria Taxa

Table 1 part B and C show the relative coverage of Cnidaria and non-Cnidaria across all three surveys. Scleractinia dominated the Cnidaria coverage in TQ4 in all years, ranging from 45.2% in 1981 to 32.4% in 1991 and 41.7% in 2019. Hydrocorals, alcyonarian soft corals, and other cnidarians decreased or disappeared over the three decades.

Coralline red algae tripled their coverage since 1991, increasing from 1.9% and 53 patches in 1981 to 3.2% and 286 patches in 2019. Coverage of *Porifera* decreased from 0.5% and 149 individuals in 1981 to 0.1% and 13 individuals in 2019.

### 3.3.3. Changes in the Relative Abundance of Hermatypic Coral Growth Forms

Table 1 part D shows the relative abundance of distinctive growth forms of hermatypic corals, including branching corals, massive corals, encrusting corals, and free-living corals, analysed according to the methodologies outlined by Mergner and Schuhmacher (1985) [5] and Reinicke et al. (2003) [24]. "Branching" corals, which encompassed all *Acropora* spp., *Stylophora pistillata*, and *Pocillopora verrucosa* decreased over the survey period from 38.3% to 28.4%. "Massive" forms included species from the genera *Porites*, *Acanthastrea*, *Coelastrea*, *Dipsastraea*, *Favites*, *Galaxea*, *Goniastrea*, *Leptastrea*, *Leptoria*, and *Lobophyllia* decreased from 6.3% in 1980 to 3% in 1991, and regained coverage to 4.3% in 2019. "Encrusting" growth forms including species from the genera *Psammocora*, *Montipora*, *Pavona*, *Echinopora*, and *Gardinoseris* decreased in coverage from 5.6% in 1981 to 3.2% in 1991, recovering to 7.3% in 2019. Free-living forms were identified within the Fungiidae family. It is worthwhile noting that the overall proportions of these growth forms remained relatively stable during the investigated period (Table 1 part D).

The aforementioned phenomenon also applies to the relative abundance of hermatypic corals, which actively participate in the construction and development of coral reefs. Hermatypic corals include all Scleractinia together with the hydrozoan corals such as *Millepora* and *Distichopora*, whereas Xeniidae and Zoanthidae represent non-reef building taxa.

### 3.3.4. Changes at the Genus and Species level

Analysing the results of all three surveys also revealed trends in the relative abundances of different coral genera and species, as shown in Figure 12B. At the genus and species level, coverage of *Acropora* spp., which was dominant in TQ4 in 1981 and 1991, decreased by nearly 50% across the three surveys. *A. hemprichii* comprised over 10.4% of the living coral coverage in 2019. The coverage of *Pocillopora verrucosa* doubled after a slight decrease in 1991 to 38.7% in 2019. Massive species, such *Porites* spp. and *Pavona* spp. sustained their shares, and were complemented by 5.3% of *Montipora* sp. *Millepora dichotoma* doubled its percentage cover, from 2.1% in 1980 to 3% in 1991 and 4.5% in 2019. In contrast, *Millepora platyphylla* disappeared after 1991. Fungiidae coral cover halved their share to less than 1%. Massive corals from the family Merulinidae have increased their

proportion of cnidarian cover coverage from 10.5% in 1980 to 14.1% in 2019, while the size or number of other taxa sighted has not changed significantly.

### 3.4. Comparison over Time 1980–2019 in Test Plot TQ4

3.4.1. Statistical Evaluation

The comparative analysis of the results from all three surveys revealed both continuity and changes within the coral community at TQ4. Information on the colony numbers and coverages of taxa between years is provided in Table S1. The analysis was to identify and highlight small variations and modifications within the coral community and establish their correlation with abiotic factors.

For the quantitative analysis, a Kolmogorov–Smirnov test was conducted to assess the normal distribution of the data for all three survey periods. However, the test results indicated that the samples did not follow a normal distribution ($p$-value (two-tailed) = 0.0001). Consequently, Friedman's test, which is suitable for non-parametric data, was performed to ascertain that all three datasets originated from the same population.

In addition, a rank abundance plot was created (Figure 13), depicting the abundance ranking of the observed species in all surveys and their respective coverage segments within the test plot in cm$^2$. In 2019, the species with the highest abundance was *Pocillopora verrucosa*, covering an area of 43.2 cm$^2$, compared to 20.9 cm$^2$ in 1980 and 16.7 cm$^2$ in 1991. In the 1980 and 1991 surveys, *Acropora* cf. "*superba*" was ranked the most abundant species, with coverage areas of 31.0 cm$^2$ and 32.5 cm$^2$, respectively. Out of the 47 identified cnidarian species, 18 species exhibited total abundances exceeding 1.00 cm$^2$, compared to 20 out of 52 in 1980 and 13 out of 51 in 1991. The 1980 plot included species with the lowest abundance, such as *Seriatopora caliendrum* with 5 cm$^2$ and *Favites rotundata* which covered 2 cm$^2$.

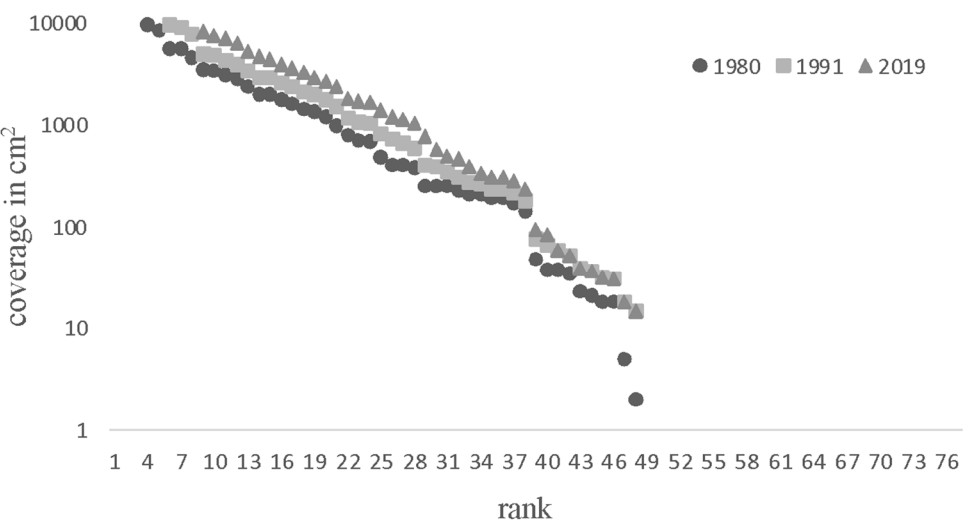

**Figure 13.** Rank abundance plots for the 1980, 1991, and 2019 surveys.

3.4.2. Changes of Coral Biocoenosis on Species Level

The interpretation of changes in living substrate data necessitates careful consideration of the methodological differences employed during each sampling event. It is crucial to assume that reidentified individual colonies in similar positions represent either the same colony or its fragments in the same location. To account for the conversion of in situ three-dimensional data into a two-dimensional map, a methodological error of 5% per monitoring event was estimated. This translates to an error of 10% for two-way comparisons.

3.4.3. Species Turnover Rates after Schoener 1983

Temporal changes in benthic composition were assessed using the Schoener (1983) method to calculate the appearance ($I_{abs}$) and disappearance ($E_{abs}$) of species, and species turnover rates between 1980–1991, 1980–2019, and 1991–2019 (Table 2).

**Table 2.** Species turnover rate at TQ4 for three different time periods ($T_{rel}$ (%/yr) after Schoener, 1983). $I_{abs}$ represents the number of appearing species, $E_{abs}$ represents the number of disappearing species, $S1$ represents the total number of species at the start of the period (1980, 1991, 1980), and $S2$ represents the total number of species at the end of the period (1991, 2019, 2019).

| | Turnover Rate | | |
|---|---|---|---|
| | 1980–1991 | 1991–2019 | 1980–2019 |
| $I_{abs}$ | 14 | 15 | 11 |
| $E_{abs}$ | 18 | 15 | 15 |
| $S1$ | 47 | 43 | 47 |
| $S2$ | 43 | 46 | 46 |
| $T_{rel}$ **(%/yr)** | 3.23 | 1.2 | 0.72 |

Taxonomic revisions were taken into account (after coralsoftheworld.org) [31] and species were named according to current taxonomic updates (e.g., after WoRMS 2023) [9]. Fungiidae, *Porites*, *Ovabunda*, and *Montipora* species were generally counted as "1 taxon present in 2019" for this calculation, due to limitations in taxonomic resolution from photo IDs. The overall coverage of these groups is similar between years with taxa inaccuracies of <5% (1980, 2.66%; 1991, 3.23%; 2019, 4.37%) and <10% of the cnidarian cover (1980, 5%; 1991, 7.64%; 2019, 4.75%; *Ovabunda*: 1980, 1.76%; 1991, 6.38%; 2019, 1.14%; *Montipora*: 1980, 0.3%; 1991, 0.08%; 2019, 0.18%; Fungiidae: 1980, 1.81%; 1991, 0.98; 2019, 0.94; *Porites*: 1980, 1.13%; 1991, 0.2%; 2019, 2.15% of cnidarian cover).

The need to count "1 taxon present in 2019" for these groups resulted in a discrepancy in the turnover rates calculated in our study, yielding a value of 3.23 for the time period 1980–1991 (see Table 2), and the findings of Reinicke et al. (2003) [24], who reported a value of 3.03.

Nevertheless, a discernible trend persists. The majority of appearing and disappearing species belonged to the genera *Acropora* and *Pocillopora* (branching corals) and the Merulinidae (massive corals), according to Table S2. Furthermore, the median values of the numbers of appearing and disappearing species during different time intervals indicate that the species turnover rate decreases with longer periods.

3.4.4. Multivariate Analysis

Results of the multivariate analysis are shown in Figures 14 and 15. Figure 14 shows the clustering of samples based on cnidarian abundance within TQ4 for the three sampling surveys, with each individual data point representing one survey (1980, 1991, and 2019). The SIMPROF analysis identified one significant cluster (1980 and 1991, Group b), with a similarity of 75.64 and one outlier (2019, Group a).

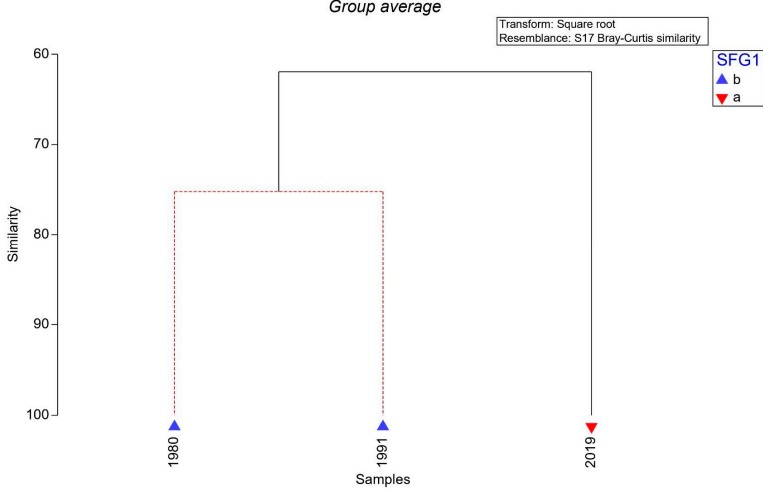

**Figure 14.** Dendrogram showing results of cluster analysis of cnidarian cover within TQ4 (standardised and square root transformed), including SIMPROF results.

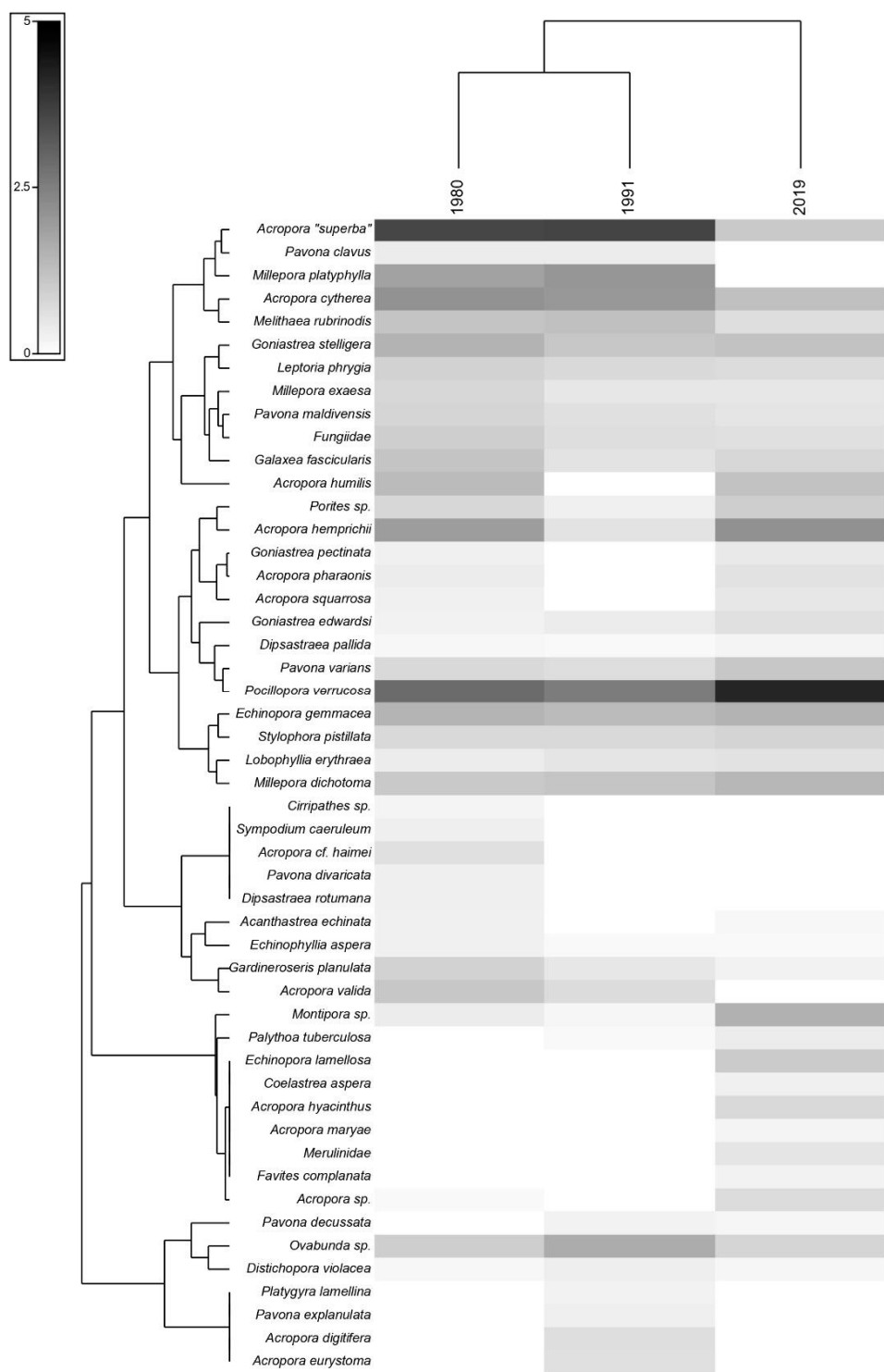

**Figure 15.** Shade plot of cnidarian cover data (standardised and square root transformed), showing the top 50 contributory species across all years and the clustering of these species (index of association) and samples (Bray–Curtis similarity).

The shade plot in Figure 15 shows the data matrices, representing the top 50 species and their respective changes in abundances through the sampling intervals.

SIMPER analysis based on square root transformed data indicated the similarity between Group b (1980 and 1991) was driven by species such as *Acropora "superba"*, *Pocillopora verrucosa*, *Acropora cytherea*, *Millepora platyphylla*, *Echinopora gemmacea*, *Goniastrea stelligera*, *Melithaea rubrinodis*, *Millepora dichotoma*, *Ovabunda* sp., *Leptoria phrygia*, and *Acropora valida*.

The dissimilarity between the composition of the coral community found within TQ4 between Group b (1980 and 1991) and outlier Group a (2019) was due to differences in the relative abundances of species. For example, coral taxa abundances that were higher in Group b compared to Group a included *Acropora "superba"*, *Millepora platyphylla*, *Acropora valida*, *Acropora cytherea*, *Melithaea rubrinodis*, and *Ovabunda* sp. Conversely, coral taxa where abundances were higher in Group a compared to Group b included *Pocillopora verrucosa*, *Montipora* sp., *Echinopora lamellosa*, *Acropora hemprichii*, *Acropora hyacinthus*, *Acropora* sp., *Acropora humilis*, *Merulinidae*, *Porites* sp., and *Pavona varians*. The full results of the SIMPER analysis are presented in Table S3.

## 4. Discussion

Hans Mergner and Helmut Schuhmacher initiated this long term study of Sudanese coral reefs in 1980, along with other early works on the Red Sea coast of Sudan [36]. In parallel with British and French researchers who were exploring the coral reefs of Sudan, their specific aim was to compare coral communities and environmental conditions along a transect from coastal to off-shore reefs, which initially also included sites at Port Sudan Lighthouse and Wingate Reef. The two latter sites were later abandoned, and the study focused solely on the Sanganeb atoll [5]. Those early studies laid the foundation for a longer-term study, without a predefined schedule for repetition. The first opportunity to repeat the study occurred 11 years later, during a German Research Foundation (DFG)-funded program to investigate fossil and recent reef facies between 1990–1995. H. Schuhmacher and his team returned to the Sanganeb atoll in 1991, to compare the plots and investigate dynamic aspects of the coral communities [24].

Since then, tropical coral reef environments have changed drastically, both within the Red Sea region, and on the global scale. Shallow tropical reefs, i.e., nearshore benthic communities, face a wide range of threats of local to worldwide significance. Human presence along tropical coasts has increased considerably alongside population growth and the expansion of coastal settlements. The Red Sea, situated in a semi-enclosed basin in a dry region, is considered one of the warmest marine ecosystems on Earth [18].

### *4.1. Net Vertical Reef Accretion*

A reduction in the net vertical reef accretion estimate in the TQ4 framework structure is detected over the past 28 years from 8–12 cm between 1991 and 2019 (2.8–4.3 mm yr$^{-1}$), compared to 25–30 cm during the earlier 11-year interval between 1980 and 1991 (22.7–27.3 mm yr$^{-1}$) (Figure 11). Following the initial period without (recorded) significant impact on the reefs, the subsequent three decades resulted in a substantial decline in net reef accretion of over 80%. The persistence of (at least) two individual long-lived colonies present in all three surveys (Figure 10A,B) provides evidence of the overall favourable conditions at the site. The reduced abundance of *Acropora* spp. and the spread of *Pocillopora verrucosa* (doubled vs. 1980, tripled vs. 1991) indicates other influences. A large proportion of the coral colonies are young (estimated to be ~3–4 years old), which suggests recolonisation after a disturbance event and reflects ongoing changes in the community.

The recorded rates of vertical reef accretion exhibit considerable variability in both recent and fossilised datasets. In high-energy reef crest environments, observed values can be less than 12 mm per year, while in more sheltered and protected (deeper) areas, the rates may reach up to 25 mm per year [37]. Fossil core data obtained from Aqaba in the northern Red Sea indicate a range of 0.67 to 1.69 mm per year over a span of 5000 years in the late Holocene. In comparison, the Sanganeb atoll displays accretion rates ranging from 1.6 to 6.0 mm per year throughout the Holocene, with higher values observed during the early to mid-Holocene era [38].

The equilibrium between reef-building calcifiers and bio-erosive organisms and processes is intricately influenced by a range of abiotic factors, including exposure and current patterns, light conditions, depth, and water temperature [39]. Notably, at the TQ4 site, during the survey period in February–March 1980, the reported mean values of wave height

reached at least 1.1 m [5]. The authors observed a non-destructive, moderate to occasionally strong current regime at TQ4, resulting in minimal deposits of fine sediments in the plot. Scuba diving was only viable on calm days. More recent data show that monthly wave heights in the region were typically below 1.0 m, primarily influenced by basin depth and the limited wind fetch distance (data from 2008–2009) [40].

In the central Red Sea, between 19–20° N, complex mixing occurs during winter due to converging wind vectors from the north and south, driven by the intense coupling of wind and sea surface temperature (SST) [41]. Severe storms can occur but are reported to be exceptionally rare. Underwater light conditions, which are generally excellent with visibility exceeding 30 m on calm days [5], were observed to turn turbid on windy days in September 1991 (GBR, personal observation). There is a change in the angle of slope and hence, the orientation of the surface within TQ4, with a 25° slope in the top left segment and a steeper 40° slope in the bottom right segment [5,24]. Despite this change in orientation and light regime, there is no apparent limitation on coral growth and development.

Bioerosion, which involves the breakdown of calcified coral skeletons, is orchestrated by various small organisms such as fungi, algae, and bacteria, as well as boring sponges, and grazers like echinoids or corallivorous fish. However, there is a lack of specific evidence regarding the occurrence of bioerosion in this particular context.

Consistent reports indicate the absence of grazers, including the sea urchins *Diadema setosum* (Leske, 1778) and *Echinometra mathaei* (Blainville, 1825), on the atoll. In contrast, observed densities of 10–20 individuals per 100 m$^2$ were noted in coastal reefs south of Souakin [11]. Furthermore, the coral-eating snail *Drupella cornus* (Röding, 1798) was not present, although it was observed on the reefs of Al-Wajh in Saudi Arabia (GBR, personal observation, 2013).

Occurrences of the crown-of-thorns (COT) starfish *Acanthaster* sp. have been documented in Sudanese reefs since the 1970s, with low abundances observed during coastal surveys conducted between 2002 and 2007 [11]. However, outbreaks have not been reported in the Sanganeb atoll. Other factors such as eutrophication and rising temperatures can expedite processes of reef deterioration by inducing coral mortality and intensifying bioerosion processes. Recent findings indicate that ocean acidification may further accelerate the dissolution of calcium carbonate structures [42].

Our discovery of a diminished rate of vertical reef accretion is indicative of larger-scale impacts on the Sanganeb atoll's coral communities. Raitsos et al. (2011) documented a sudden warming trend in the central Red Sea post-1994, subsequent to the TQ4 survey conducted in 1991 [17]. In the mid-1990s, the SST in the Red Sea abruptly and significantly increased by 0.7 °C [17], in advance of the first reported severe mass global coral bleaching event in 1997/1998 [43]. The intensification and increased frequency of El Niño events, attributed to global warming [44], has resulted in repeated mass coral bleaching events around the world [45,46]. An SST-related decline in coral growth of *Diploastrea heliopora* by −30% has been observed since 1998 [47].

In the course of the widespread warming event of 1997/1998, coral bleaching was observed in northern Sudan reefs at Dungonab [48] and on Saudi Arabian reefs at similar latitudes in August 1998 [49]. Widespread coral bleaching was observed by one of the commercial dive operators working in Sudan in late 2010 (R. Klaus pers. communication, MSY Elegante). Significant amounts of dead coral and coral rubble were observed at the Sanganeb atoll during a survey at other sites in 2012 (R. Klaus pers. observation, present study). Coral bleaching was reported from other sites in the central Red Sea region in 2010 (10–11 DHW) [50]. Further heat anomalies exceeding +1.0 °C above mean summer T$_{max}$ values of SSTs were recorded, e.g., from stress bands in *Porites* coral cores from Saudi Arabian reefs in the region during the 2015/2016 global bleaching event [51,52]. For the first decade of this century, an increase in SST by 0.7 °C was reported from satellite data for the whole Red Sea [17]. The Port Sudan region has experienced warming by 0.4–0.45 °C decade$^{-1}$, and approx. 2.0 °C over 40 years [18]. Decadal trends in the Red Sea mean temperature values (28–29 °C at 15 m depth around Sanganeb, 19.5° N, data from

1958–2017, [53]) report an increase of $0.045 \pm 0.016\,°\text{C}\ \text{decade}^{-1}$. Maximum temperatures ($T_{max}$, 1982–2015) indicate mean values of 31–32 °C for the Port Sudan region, with a mean increase in $T_{max}$ of $0.17\,°\text{C} +/- 0.07\,°\text{C}\ \text{decade}^{-1}$ in the entire Red Sea.

### 4.2. The Status of TQ4 in 2019

In March 1980, the water temperature for TQ4 at the Sanganeb atoll was recorded as 26.0 °C [5]. Analysis of the collective observations pertaining to the conditions influencing coral community development at TQ4 leads us to the conclusion that the observed reduction in reef accretion during the 2019 monitoring period, in comparison to the previous 11 and 28 years, is likely a result of regional-scale events such as marine heatwaves and coral bleaching-associated mortality. These events have implications for both individual colony life histories and the overall composition of the community.

While certain massive species colonies have endured for over 39 years, there has been a consistent shift in community composition, marked by the abundant recruitment of branching coral species following these frequent disturbance events. However, it is essential to acknowledge that these observations present an inclusive estimate, and the influence of other potential factors cannot be independently evaluated.

The TQ4 study plot is characterised by the prevalence of branching growth forms among scleractinians, with 47 identified species covering 41.1% of the investigated area. The dominant species within the plot are *Pocillopora verrucosa* and *Acropora hemprichii*, accounting for over 40% of the total area colonised by cnidarians. Notably, *Acropora "superba"* and *P. verrucosa* had already emerged as dominant species in the plot in 1980 and 1991, respectively. Furthermore, various *Acropora* species contributed an additional 12.9% to the overall coverage of cnidarians, with a total of 21 colonies.

### 4.3. Taxonomic Consistency

To ensure consistency across the survey intervals, there were several species that had to be reclassified according to recent systematic revisions as presented in Table 3. The dominant *Acropora* species identified in the 1980 and 1991 surveys was referred to as *A. superba* (Klunzinger, 1879) with explicit reservation, as it was never again reported after the original description [5,32]. Indeed, one of Klunzinger's own remarks questions his original label of origin as being from the Red Sea, instead suggesting it may be from the West Indies, based on an epizoic gastropod species on the colony. As the type specimen is lost (A. Baird, pers. communication), that species' status remains dubious. For future reference, we hereby present photographs of the site and a colony close-up (Figure 16A,B), as well as reference specimens collected by Mergner's team in 1980 (Figure 17A–D). In this paper, we refer to the species observed as *Acropora "superba"* (Klz., sensu Mergner and Schuhmacher, 1985 [5]).

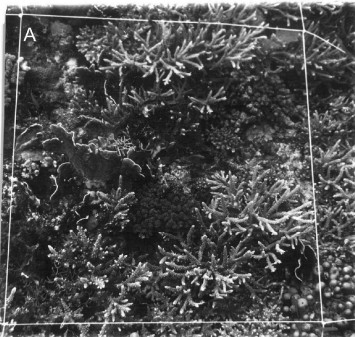 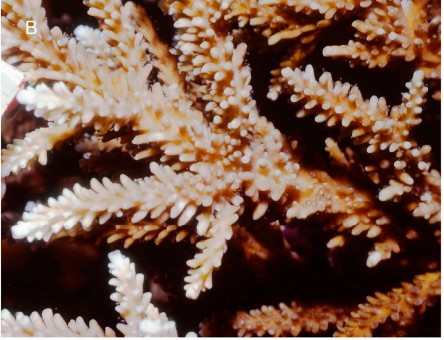

**Figure 16.** Reference documentation for the identification of *Acropora "superba"* (Klz., sensu M and S, 1985) from the survey plot TQ4 at the Sanganeb atoll. (**A**) Showing detail of b/w photographic documentation of TQ4, sector "V-d" (1 m × 1 m), with colonies of dominant *A. "superba"*, *Pocillopora verrucosa*, and *Millepora platyphylla* (*i.a.*); (**B**) close-up colour slide 1:3 of live *A. "superba"* (photo credit and identification: H. Schuhmacher, 1980).

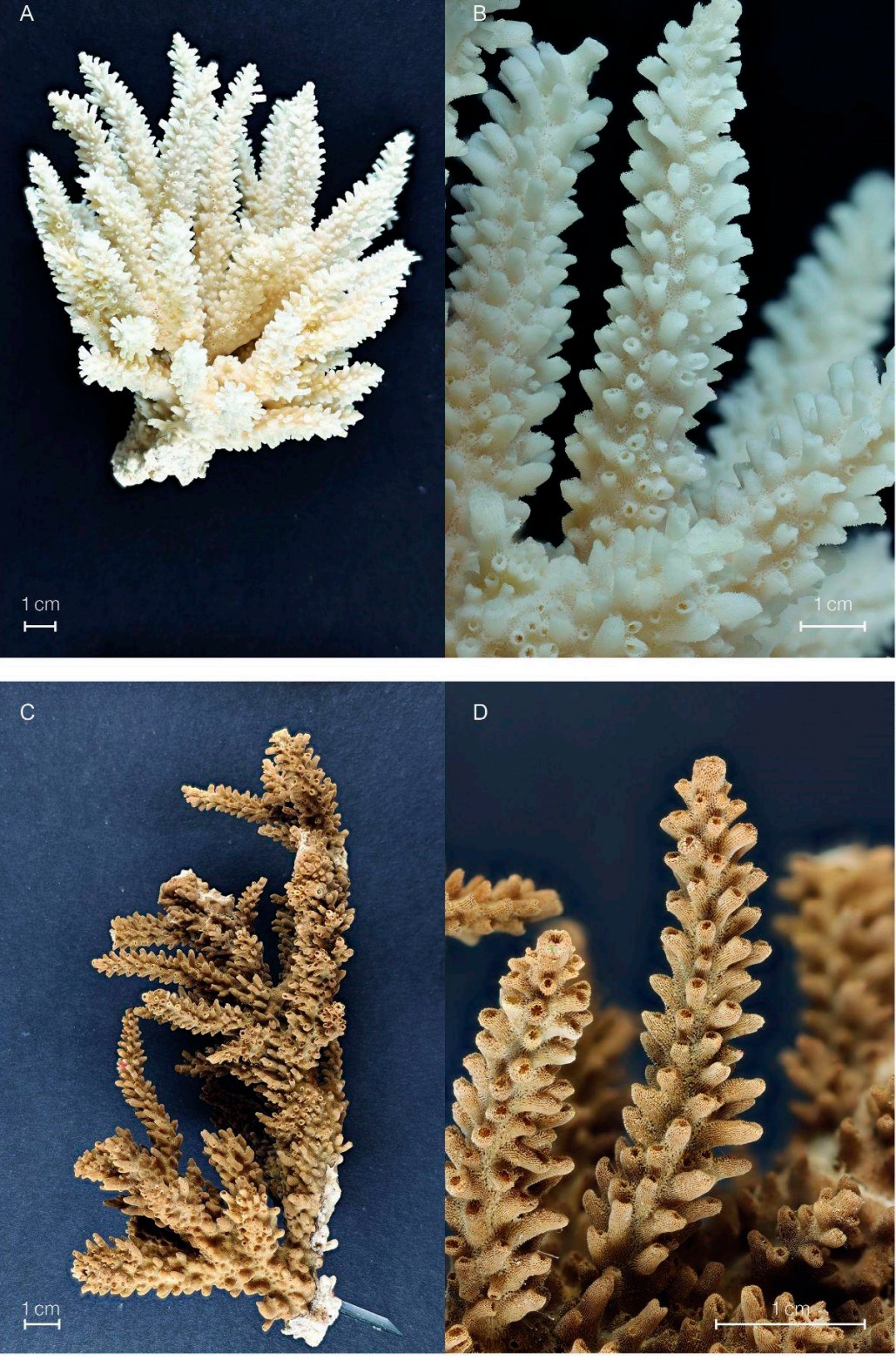

**Figure 17.** Reference specimens of *Acropora "superba"* from TQ4 in 1980. (**A**): GOM-HS-coll. #226: ID noted in H. Schuhmacher's handwriting. (**B**) Detail of #226; (**C**) GOM-HS-coll. #228: ID noted in H. Schuhmacher's handwriting. (**D**) Detail of #228, all scale bars 1 cm (photo credit: T. Moritz).

**Table 3.** Former taxon names used in the 1980 and 1991 surveys (left column) and their updated (2023) names refer to the literature and World Register of Marine Species (WoRMS) on the right side. * Not listed in WoRMS.

| Former Names in the 1980 and 1991 Surveys | Updated Taxa Names [9] |
| --- | --- |
| *Acropora corymbosa* (Lamarck, 1816) | *Acropora cytherea* (Dana, 1846) |
| *Acropora squarrosa* (Ehrenberg, 1834) | *Acropora maryae* * Veron, 2000 |
| *Acropora superba* (Klunzinger, 1879) | *Acropora "superba"* * (Klunzinger, 1879, sensu [5]) |
| *Acropora variabilis* (Klunzinger, 1879) | *Acropora valida* (Dana, 1846) |
| *Clathraria rubrinodis* (Gray, 1859) | *Melithaea rubrinodis* (Gray, 1859) |
| *Favia favus* (Forskål, 1775) | *Dipsastraea favus* (Forskål, 1775) |
| *Favia rotumana* (Gardiner, 1899) | *Dipsastraea rotumana* (Gardiner, 1899) |
| *Favia speciosa* (Dana, 1846) | *Dipsastraea speciosa* (Dana, 1846) |
| *Favia stelligera* (Dana, 1846) | *Goniastrea stelligera* (Dana, 1834) |
| *Favia pallida* (Dana, 1846) | *Dipsastraea pallida* (Dana, 1846) |
| *Goniastrea aspera* Verrill, 1866 | *Coelastrea aspera* (Verrill, 1866) |
| *Goniopora minor* Crossland, 1952 | *Goniopora pedunculata* Quoy and Gaimard, 1833 |
| *Symphyllia erythraea* (Klunzinger, 1879) | *Lobophyllia erythraea* (Klunzinger, 1879) |
| *Fungia scutaria* Lamarck, 1801 | *Lobactis scutaria* (Lamarck, 1801) |
| *Fungia horrida* Dana, 1846 | *Danafungia horrida* (Dana, 1846) |
| *Fungia klunzingeri* Döderlein, 1901 | *Danafungia horrida* (Dana, 1846) |
| *Fungia echinata* (Pallas, 1766) | *Ctenactis echinata* (Pallas, 1766) |
| *Xenia* spp. Lamarck, 1816 | *Ovabunda* spp. Alderslade, 2001 |

Among the few nominal arborescent *Acropora* species described from different type locations in the Red Sea, none appear to be in agreement with the two reference specimens collected in Figure 18 (A. Baird, pers. communication). The closest affinity to established species was suggested with *A. microphthalma* (Verrill, 1869) from the Ryukyu archipelago (Japan). Other species described from Indian Ocean locations did not match the Sanganeb atoll sample material and observations. For consistency with the earlier publications, we provisionally use the name as *A. "superba"* sensu Mergner and Schuhmacher (1985) here until the future definition of the species' status.

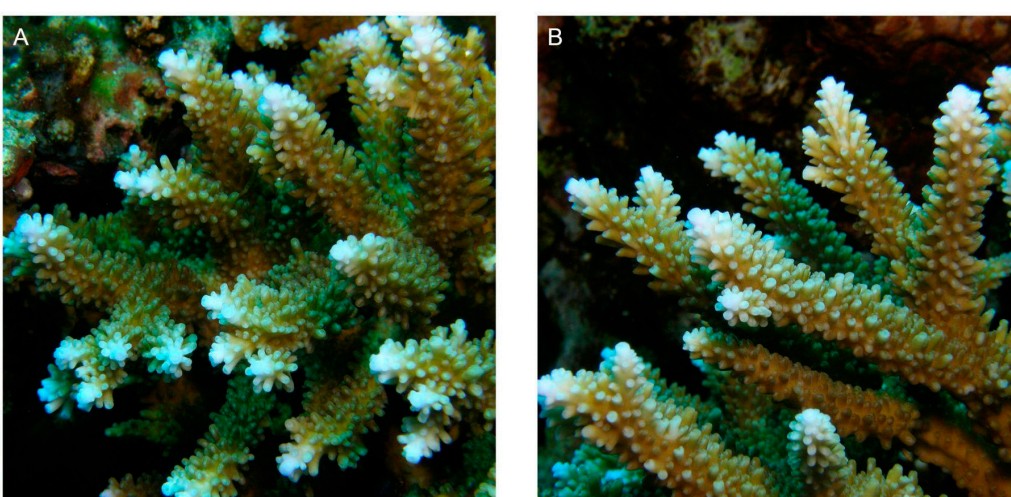

**Figure 18.** Reference documentation for the identification of *Acropora "superba"* (Klz., sensu M and S, 1985) in the survey plot TQ4 at the Sanganeb atoll in 2019. (**A**,**B**) Colony details in the plot sectors "IV/V-c/d" (2 m × 2 m) showing the dominant, but yet unnamed *Acropora* species (photo credit: J. Höhn).

*Acropora maryae* Veron, 2000, is not accepted as a valid species and is considered a junior synonym of *A. squarrosa* (Ehrenberg, 1834), according to WoRMS (2023) [9]. Veron (2000) [13] describes the extended tubular structure of the axial corallites in *A. maryae* to

differ from *A. squarrosa* whose axial corallites are also dome-shaped and have thick walls, but are very short and close to the main branches. Thus, as the colony found in the plot phenotypically differs from other *A. squarrosa* colonies, it was referred to as *A. maryae* in this study (see Figure 19A,B and Table 3).

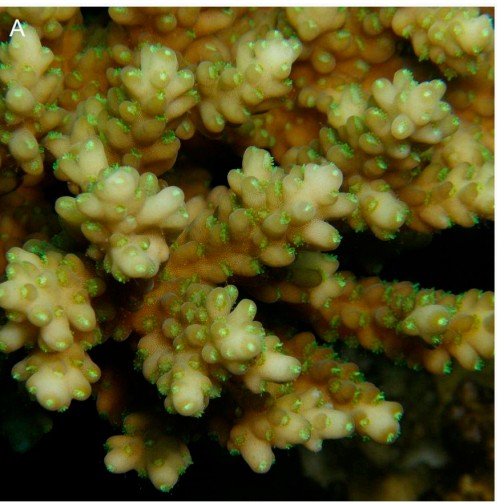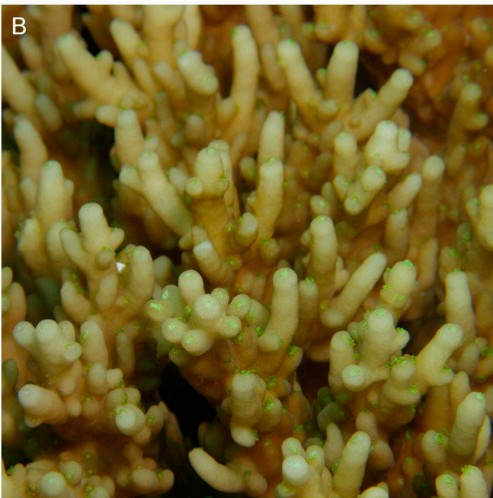

**Figure 19.** Photographs of (**A**) *Acropora squarrosa* (Ehrenberg, 1834); (**B**) *Acropora maryae* Veron, 2000 colonies from TQ4, Sanganeb atoll (Sudan) (photo credits: J. Höhn, 2019).

The other species that were reclassified according to recent systematic revisions are presented in Table 3.

### 4.4. Indicator Species and Bio-Physiographic Zone

Indicator species were identified, based on both quantitative data and visual prominence within the plot, as well as their substantial number of colonies and coverage area, by extracting data from Table S1. The selected species are visually evident in Figures 7–9. These indicator species, in conjunction with the delineation of bio-physiographic zones, enabled the differentiation of areas characterised by notable colonisation patterns, surpassing other species in terms of prevalence. The classification of these zones relies on the dominance of specific colonisers rather than exclusively on the geological structural properties, as detailed by Mergner and Schuhmacher (1974) [19]. Consequently, this specific bio-physiographic zone is renamed based on the relative shares as a *Pocillopora verrucosa–Acropora* zone.

The overall living cover of these corals has experienced a slight decline, decreasing from 56% in 1980 to 48% in 2019, even though there was significant mortality observed in 2012 following earlier bleaching events (R. Klaus, personal observation).

Studies have shown that the distribution of coral communities varies based on the horizontal gradient of wind and wave exposure [54]. Juvenile *Montipora* colonies, for example, tend to be more abundant in reefs with heavy wave exposure and high calcareous algae cover [55]. Over time, both coralline algae and *Montipora* colonies have increased in coverage and colony numbers. Encrusting species, in general, are more common in high-energy locations due to their mechanical advantages.

Quantitative analysis supports these observations. The distribution of certain coral colonies, such as *Millepora* species and encrusting Scleractinia like *Echinopora* and *Montipora*, follows a gradient corresponding to water flow orientation. *Millepora* colonies in particular tend to align their broadsides against the main direction of water movement. This behaviour makes them a reliable indicator of water flow orientation [41,56–59].

### 4.5. Dynamic Process Evaluation

The dynamic evaluation of coral species involves assessing and analysing various factors that influence the health, resilience, and survival of a coral reef community. It

is considered to be a multidimensional approach that reflects both natural and human-influenced processes affecting the coral population over time. Here we attempt to analyse some dynamic processes of coral on the species level (Figure 20).

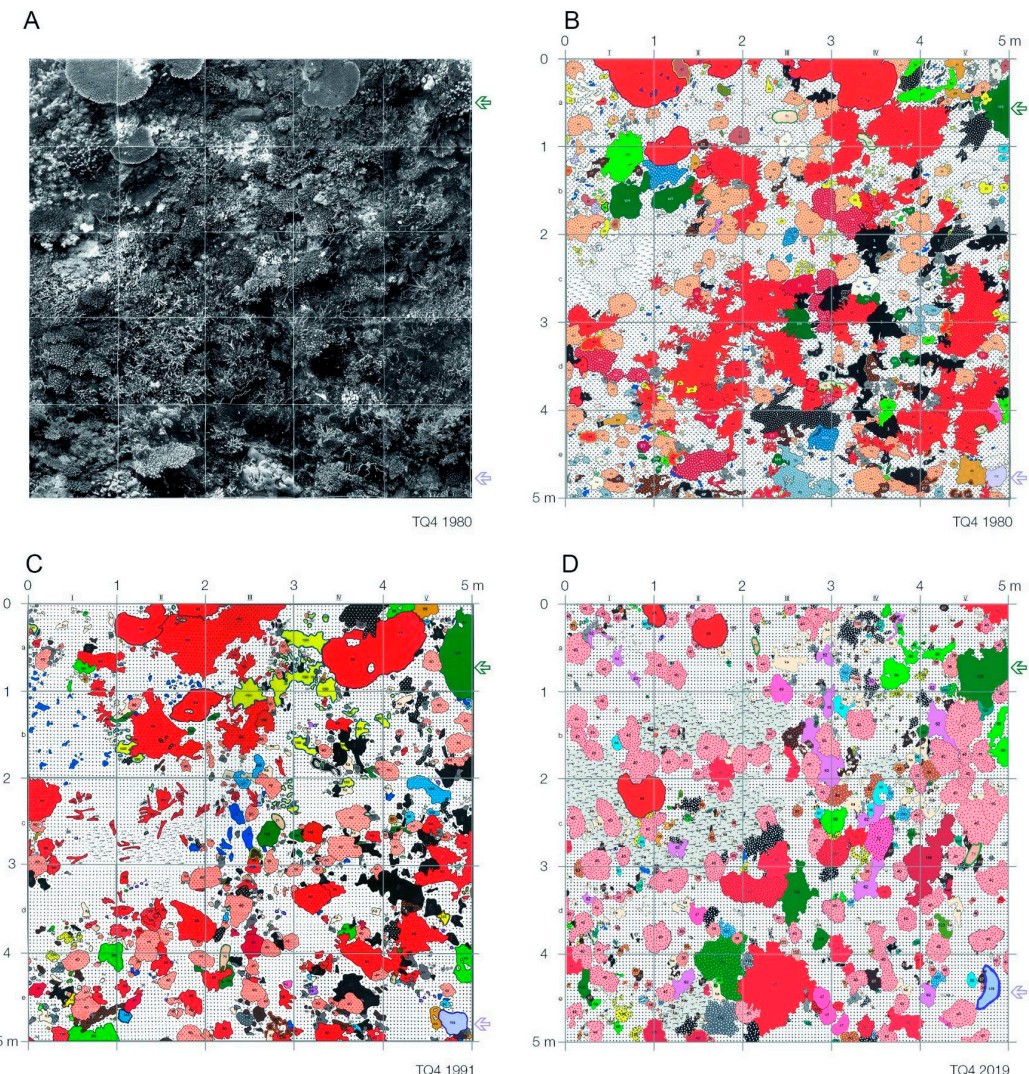

**Figure 20.** A total of 39 years of reef life encapsulated within plots of the coral community at test plot TQ4, Sanganeb atoll (Sudan); (**A**,**B**) show the photomosaic and coral community map from the initial survey in 1980. The colour codes of taxa are in agreement with the subsequent plots; (**C**) plot TQ4 of 1991 and (**D**) of 2019—see legend in Figure 9. The comparison reveals persistence at the individual and assemblage levels. A certain degree of consistency as well as dynamic changes in the community are reflected on the background of a stable geomorphological setup. While 86 colonies of *Pocillopora verrucosa* indicate recovery after a recent disturbance event, individual colonies demonstrate suitable conditions over 39 years. Arrows indicate *Echinopora gemmacea* in the coordinate Va (top right) and *Lobophytum erythraea* in Ve (bottom right) of the plots. Also, preferred positions of *Millepora* spp. (black) appear stable over almost four decades.

*Pocillopora verrucosa* is often prevalent in strong surf-exposed locations. In TQ4, the previously dominant *Acropora* species have now been replaced by *Pocillopora verrucosa*. The *Pocillopora verrucosa* colonies recorded within TQ4 2019 are generally the same size, suggesting quasi-simultaneous settlement. Tortolero-Langarica et al. (2017) provided an average growth rate of 3.3 cm per year and an average colony area of 502 cm$^2$, so it can be estimated that the *P. verrucosa* colonies in TQ4 are approximately 4 years old [60]. These observations might indicate the past third global mass bleaching event in 2015/2016 as the

origin, which was documented by NOAA in the central Saudi Arabian Red Sea, and may have also impacted the Sanganeb atoll [39,51,61,62].

Since the mid-20th century, stress-tolerant and fast-growing reef-building corals, along with competitive *Millepora* colonies, have progressively replaced the once dominant *Acropora* corals in shallow-water reef systems across the Caribbean [40]. A similar shift has been observed in the northern Great Barrier Reefs, where *Pocillopora* replaced *Acropora* as the dominant coral genus in 2021 [63]. *Pocillopora* can become dominant in highly disturbed environments, either temporarily or as a long-term shift. Bowden-Kerby (2023) outlines a progressive shift observed in Fiji sites from *Acropora* dominance to an intermediate stage characterised by *Pocillopora*, and ultimately to *Porites* prevalence. These ecosystems serve as invaluable case studies, providing essential insights. The survival of broadcast spawning coral genera, like *Acropora*, depends on acquiring zooxanthellae within a short period of time after settlement to avoid mortality, while brooding coral genera such as *Pocillopora* sustain longer-lived larvae through their ability to produce food. Coral reefs lacking up-current sources of heat-adapted larvae are more vulnerable to phase shifts towards dominance by brooding species, as observed in the test plot [64]. In historical contrast, *Millepora dichotoma* and *Pocillopora verrucosa* have been discovered in Pleistocene reef crests in the Red Sea [65]. The abundance of *Millepora* in the Pleistocene correlates with the high abundance of *Pocillopora* [66]. Hydrocorals, including *Millepora*, are opportunistic species with high reproductive rates and rapid growth, and they are avoided by the corallivorous predator *Acanthaster* sp. *planci* [67]. The presence of *Millepora* fans indicates exposure to either waves or currents [66]. *Millepora dichotoma* is typically the most prevalent milleporid coral in modern Red Sea reefs [54,66,68]. The sheet tree-shaped *Millepora platyphylla* is particularly susceptible to wave-induced breaking, which may explain its absence in the 2019 investigation [67]. Demographic studies by Lenihan et al. (2011) from 2005 to 2007 suggest that *Acropora* was more negatively affected by corallivory impacts than *Pocillopora*, providing a possible explanation for the observed changes in coral community composition in Moorea [69]. There, *Pocillopora* exhibits excellent adaptation to partial predation by snails [69,70]. Pocilloporid corals often display adaptive life strategies, attributed to their reproductive mode as breeders, rapid growth rates, and small size (5–8 cm) until sexual maturity [55,63,71].

*4.6. Long-Term Shifting Community*

The uni- and multivariate analyses of changes in the community structures recorded in the three surveys of TQ4 indicate some development. The increase of Simpson's dominance index ($\lambda$) in TQ4 from 1980 and 1991 ($\lambda$ = 0.1093 and 0.1472, respectively) to 2019 ($\lambda$ = 1.754) confirms the observed shift in the dominance of species in the community. Intermediate values of Pielou's evenness (J') indicate moderate dominance of the most abundant species like *Acropora* "*superba*", *Pocillopora verrucosa*, *A. cytherea*, *Millepora platyphylla*, and *Echinopora gemmacea*, as well as *Montipora* sp., in 2019 (compare Table 1, Figures 9 and 12B).

A comparison of diversity measures from seven reef locations along the mainland coast revealed slightly differing patterns regarding species richness and community structures. However, these results need careful consideration as the applied methodologies were not the same. Based on the survey of seven Sudanese fringing reef sites spanning 300 km of coastline from Oseif to Suakin (2008–2011), Ali and Elhag (2016) [72] assessed coral community habitats at 1–3, 5–7, and 10–12 metre depths for the species composition and community measures of (i.a.) Shannon–Wiener (H') and Simpson's (D) diversity. Based on three-point intercept and belt transects of 50 m length per depth, the highest value of species richness (S' = 38) at Oseif was exceeded in the Sanganeb offshore location of TQ4 by 23–36%. The overall species count of 136 in the study, however, might indicate that the chosen sample sizes may have underrepresented the real local diversity, as the values of TQ4 do not represent the entire Sanganeb atoll, as can be seen in the differences among TQs 1–4 [24]. The richness reported from the mainland coast includes many species also recorded in TQ4 in the Sanganeb. The Shannon diversity (H') values detected in that

study were highest at a 10–12 metre depth (like TQ4) and mostly ranged between 4.0 and 5.0, while the H' measures in TQ4 ranged from 3.56 to 3.95, and were thus a little lower. Evenness values could not be compared.

The cluster analysis of cnidarian cover within TQ4 (Figure 14) and the shade plot of cnidarian cover data observed in the three TQ4 surveys (Figure 15) also demonstrate shifting in the taxonomic dominance of species in the test plot over the period of 39 years. In the clustering, Group b (1980 and 1991) had higher abundances of *Acropora* "*superba*" and *Millepora platyphylla*, but also *Acropora valida*, *Ovabunda* sp., and *Acropora cytherea*, among others, while Group a (2019) had higher abundances of *Pocillopora verrucosa*, *Montipora* sp. and *Acropora* "*superba*", *Echinopora lamellosa*, and *Acropora hemprichii*, among others.

Intermediate Disturbance Hypothesis

General ideas about disturbance events representing perturbation to communities were well established in the mid 20th century, recognising peak species diversity at intermediate impact levels [73]. Initially coined by Connell (1978) [74], and later applied to coral reefs by Connell (1997) [75], the intermediate disturbance hypothesis (IDH) offers valuable insights into coral reef dynamics. The hypothesis suggests that intermediate disturbance levels promote the coexistence of different species strategies and fugitive species within ecosystems. Our study contextualises this hypothesis within the local and regional off-shore position of the Sanganeb reefs, allowing land-based sources of pollution and extensive anthropogenic impacts to be discounted, considering factors such as wave surge, temporal bleaching, and temperature fluctuations. While lacking detailed regular local disturbance observations over time, our findings indicate that diversity has been maintained comparably to the 1980 baseline, despite known disturbance events in 1997, 2010, and 2015. These observations generally align with the principles of the IDH, particularly regarding the assemblage of long-lived and fugitive species. However, shifts in community composition since 1991, such as changes in the abundance of certain species and episodic recruitment events, may reflect ongoing fluctuations in disturbance parameters. The exclusion of some components, like echinoderm erosion, complicates the interpretation, highlighting the need for further testing and additional component observations over shorter time intervals. Our results appear in line with the IDH, but further research is required to refine the understanding and compare long-term observations from other reef locations, such as the Heron Island reefs in the Great Barrier Reef (Connell et al., 1997) [75]. Even though our study recalculated species dynamics for the numbers and sizes of colonies, and species coverage following the approach of Reinicke et al. (2003), we found that the resolution of detected dynamics in reef development over the extended time period of 28 years is not accurate enough to allow robust inferences in this study (see Table S4). Dynamics of increase (in species abundance) encompass growth, recruitment, and fragmentation, while decreasing dynamics pertain to colony retreat and loss. However, dynamics indicated that shorter periods (1980–1991) appear to exhibit more fluctuations in occurrence, disappearance, and decline, while longer periods (1980–2019 and 1991–2019) show a higher prevalence of stability and increase. Hence, these data were understood to represent continuous phases of colony formation.

## 5. Conclusions

Coral reef communities reflect ongoing changes in environmental conditions on local, regional, and global scales. Rising ocean temperatures are placing reef-building corals under significant stress beyond their temperature optimum, leading to compromised physical performance and an increased risk of bleaching [76]. Initially, it was believed that corals in the Red Sea were only minimally threatened by climate change due to their acclimation to the already elevated temperatures. While Red Sea scleractinian corals can withstand exceptionally high temperatures, they are continuously approaching their thermal limits [3,43].

In summary, the present study reveals a substantial stagnation in net vertical reef accretion rates (0.28–0.42 cm/yr) over the past 28 years (1991–2019) based on the TQ4 framework structure, as compared to the preceding 11 years period with 2.27 to 2.72 cm per annum between 1980 and 1991. This decline of over 80% is framed by the parallel persistence of long-lived colonies of *Echinopora gemmacea* and *Lobophyllia erythrea*, reflecting the overall favourable conditions. A shift in the coral species community composition dominance was evident, with a decrease in *Acropora* spp. coverage and an increase in *Pocillopora verrucosa* colony numbers and coverage. While various abiotic and ecological factors may have contributed to this shift, the research suggests that larger scale impacts, such as marine heatwaves and mass coral bleaching events in the central Red Sea, have played the most significant role in shaping the observed trends in coral growth and health.

**6. Outlook**

Hans Mergner† and Helmut Schuhmacher† deserve special merit for their foresight as the founders of this long-term study in 1980. A collection of reference specimens, b/w as well as colour slides, and plenty of documentation materials from the 1980 and 1991 studies are archived at the OMG. Available historic and recent documentation and results of this ongoing survey are available to be considered and included in the establishment of a long-term coral reef monitoring scheme within the developing Marine Park management efforts. A resurvey and analysis of further changes will help to report about dynamic developments of coral communities in the Sudanese central Red Sea.

**Supplementary Materials:** The following supporting information can be downloaded at: https://www.mdpi.com/article/10.3390/d16070379/s1, Table S1: Supplement table of the number of colonies, % of test area and % of cnidarian cover of all taxa and genera for all three survey years; Table S2: Supplement table including the change percentage in number of colonies and species coverage for all taxa and genera for all time three time intervals; Table S3: SIMPER analysis; Table S4a: Overview and summary of dynamic processes for the number of colonies, species coverage and change of mean colony size for all three time intervals and their conclusive valuation; Table S4b: Summarized results of community changes on the species level during the investigation period (1980–2019) in TQ4; Table S4c: Single and merged dynamic processes for the number of colonies, species coverage and change of mean colony size for all three time intervals and their conclusive valuation.

**Author Contributions:** Conceptualization, G.B.R.; methodology, G.B.R., S.A., R.K., and J.H.; software, G.G.; validation, G.G., S.A., and G.B.R.; formal analysis, S.A., G.B.R., R.K., and G.G.; investigation, S.A., G.B.R., R.K., and J.H.; data curation, S.A. and G.B.R.; writing—original draft preparation, S.A. and G.B.R.; writing—review and editing, G.B.R., R.K., O.S.S., and G.G.; visualisation, S.A. and J.H.; project administration, G.B.R.; funding acquisition, G.B.R.; resources, G.B.R. All authors have read and agreed to the published version of the manuscript.

**Funding:** This research: in particular the excursion to recover the 1980 and 1991 monitoring sites at the Sanganeb atoll, was initiated by Martin Zuschin at the University of Vienna (Austria) within the project "Biogeography and ecosystem stability of Red Sea coral reefs: A Pleistocene-Recent comparison", and was partly funded by the Austrian Science Fonds [FWF project P31592-B25].

**Institutional Review Board Statement:** Not applicable.

**Data Availability Statement:** All primary data are assembled in Table S1. Original b/w negatives, working field notes and documentation, as well as a reference collection of Scleractinia and colour slides of specimens recorded and identified in 1980, are available in the archive of the Ocean Museum Germany (OMG). Further, photo-documentation of the 1991 field work and reference photographs of individual colonies and close-up details in the plot surveys in 2019 and data presented in this study are available on request through the corresponding author and are curated by the second author, GBR, at the OMG. The data are not publicly available due to ongoing consultation to transfer data and include the survey documentation of the long-term coral reef monitoring scheme within the developing management efforts of the Sanganeb Marine Park Administration in Port Sudan.

**Acknowledgments:** We thank Martin Zuschin, University of Vienna, Austria, who initiated the follow-up survey of the test plots 39 years after their installation. Cpt. Lorenzo Segalini and his crew

onboard the MS "Don Questo" thoroughly navigated the Sudan off-shore reefs and supported the scuba diving and recovery of the old plots. Graphic layout design of the Figures 1, 11, 17 and 20 was assisted by Thomas Korth, and specimen photographs in Figure 17 were provided by Timo Moritz, both at the OMG. We further thank three colleagues who offered their expertise in scleractinian taxonomy to verify our photo IDs; Andrew Baird, Townsville (Australia), Bert W. Hoeksema, *naturalis*, Leiden (The Netherlands), and Bernhard Riegl, Nova Southeastern University, Dania Beach (United States)—their feedback and advice is explicitly acknowledged. Sebastian Schmidt-Roach, Bert W. Hoeksema and two anonymous reviewers provided valuable comments and stimulating suggestions to elaborate the manuscript.

**Conflicts of Interest:** Author Johannes Höhn runs the company Coralaxy in Bentwisch (Germany). The remaining authors declare that the research was conducted in the absence of any commercial or financial relationships that could be construed as a potential conflict of interest". The funders had no role in the design of the study; in the collection, analyses, or interpretation of data; in the writing of the manuscript, or in the decision to publish the results.

## Appendix A

List of species (Cnidaria) present in TQ4 on the Sanganeb atoll in 2019. Numbers refer to numerical species codes used in previous studies [5,24].

**HYDROZOA HYDROIDEA**
Milleporidae
*1 Millepora dichotoma* Forskål
*2 Millepora exaesa* Forskål
Stylasteridae
*4 Distichopora violacea* (Pallas)
**ANTHOZOA, OCTOCORALLIA**
Xeniidae
*135 Ovabunda* sp.
Cladiellidae
*147 Klyxum* sp.
Melithaeidae
*32 Melithaea rubrinodis* (Gray)
**ANTHOZOA, HEXACORALLIA**
SCLERACTINIA
Psammocoridae
*156 Psammocora profundacella* Gardiner
Astrocoeniidae
*35 Stylocoeniella armata* (Ehrenberg)
Pocilloporidae
*36 Stylophora pistillata* (Esper)
*40 Pocillopora verrucosa* (Ellis and Solander)
Acroporidae
*43 Acropora cytherea* (Dana)
*45 Acropora hemprichii* (Ehrenberg)
*46 Acropora humilis* (Dana)
*47 Acropora hyacinthus* (Dana)
*48 Acropora pharaonis* (Milne Edwards)
*49 Acropora squarrosa* (Ehrenberg)
*50 Acropora "superba"* (Klunzinger)
*51 Acropora valida* (Dana)
*52 Acropora* sp.
*149 Acropora maryae* Veron
*58 Montipora stilosa* (Ehrenberg)
*62 Montipora* sp.
Agariciidae
*65 Pavona explanulata* (Lamarck)
*66 Pavona maldivensis* (Gardiner)
*67 Pavona varians* (Verrill)
*140 Pavona decussata* (Dana)
*69 Gardineroseris planulata* (Dana)
*152 Leptoseris* sp.

Coscinaraeidae
*70 Coscinaraea monile* (Forskål)

Fungiidae
*71 Ctenactis echinata* (Pallas)
*74 Fungia (Lobactis) scutaria* (Lamarck)
*155 Fungia* sp.
Poritidae
*83 Porites* sp.
Merulinidae
*87 Dipsastraea pallida* (Dana)
*89 Dipsastraea speciosa* (Dana)
*90 Goniastrea stelligera* (Dana)
*91 Favites complanata* (Ehrenberg)
*94 Favites pentagona (Esper)*
*96 Goniastrea edwardsi* Chevalier
*97 Goniastrea pectinata* (Ehrenberg)
*100 Leptoria phrygia* (Ellis and Solander)
*107 Cyphastrea microphthalma* (Lamarck)
*109 Echinopora gemmacea* (Lamarck)
*111 Echinopora lamellosa* (Esper)
*150 Merulinidae/Faviidae*
*153 Coelastrea aspera (Verill)*
Euphylliidae
*113 Galaxea fascicularis* (Linnaeus)
Leptastreidae
*104 Leptastrea purpurea* (Dana)
Lobophylliidae
*118 Acanthastrea echinata* (Dana)
*119 Lobophyllia erythraea* (Klunzinger)
*121 Echinophyllia aspera* (Ellis and Solander)
ZOANTHARIA
Zoanthidae
*123 Palythoa tuberculosa* (Esper)
*148 Zoanthus* sp.

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
