# Peer review of "Red Sea Coral Reef Monitoring Site in Sudan after 39 Years Reveals Stagnant Reef Growth, Continuity and Change"

_diversity, doi:10.3390/d16070379_

Round 1

Reviewer 1 Report

Comments and Suggestions for Authors

General

This study presents results of repeated photo quadrat surveys conducted on the Sanganeb atoll, initially surveyed in 1980 and revisited in 1991 and 2019. The surveys found that the overall benthic community structures remained relatively consistent over the 39-year period. Some species like Echinopora gemmacea and Lobophyllia erythraea were recorded in the exact positions as in the previous surveys, and there were changes in the dominance of Acropora and Pocillopora spp. Other changes were observed in dead substrate patterns, hydrozoan, and soft coral living coverage, as well as in the accompanying fauna. The study also notes that net reef growth in the test plot declined by about 80% between the 1980s and the period from 1991 to 2019, which in my opinion, is the most interesting results and should be discussed further. The study is valuable given the extended time period between surveys (39 years), and the authors should be commended on how much work has been completed,, but it needs extensive revision to make it more concise, less reliant on earlier references for method descriptions, and better highlight the interesting results and how they relate to the environmental changes within the red sea. It would also benefit from fewer figures and replacing some obscure/unusual terms. Specific comments below:

Abstract

Line 3 – consider adding distance to provide better spatial context

Consider stating the dimensions of the photo plot i.e. 5m x 5m

Dominant “genera” rather than presence

Replace the term “shares” with proportion”

Unclear what “certain development on species level” means? Pls be more specific

What changes were observed to dead substrate etc. PLs add detail

The reference to TQ4 at this stage has no meaning to readers. Suggest referring to it as the study site or similar until its defined.

The result on Net reef growth is interesting. In my opinion this should be mentioned earlier in abstract

Introduction

Line 3 – reference or describe the conditions that support this claim

Page 2 – Abstract claims UNESCO listing in 2017 and here states 2016. Correct

Page 2 – what is rigid fishing, Please define

Page 2 – what have been the increase in SST temperatures at this study site. Consider using NOAA SST data to quantify the increase in temp at the study site

Methods

Page 4 – Consider replacing word “Notorious” with a more descriptive term such as “obvious”

Page 4 – What caused the partial burying? Sediment or coral rubble?

Page 4 – unclear why only one site selected? The changes described above would all constitute important community level changes that would be interesting to document? Based on these reasons its unclear why these sites were not analysed?

Page 4 – Paragraph commencing with “The 2019 survey….” Needs rewriting to improve clarity. Unclear why this site alone was selected and why “local impacts” would be avoided if they were typical of the reef overall?

Page 4 – Consider quantifying typical wave heights and current strengths.

Page 4 – Unclear what “lower in 1980 and 1991….” Refers to? Lower then when? 2019 surveys?

Page 5 – Unclear what “time-lapse” refers to? If this is photos taken at regular intervals state the time i.e. 1 second time-lapse. Paragraph needs editing to make clearer.

Page 5 – consider adding reference images to supplementary files

Page 6 – Consider defining the term “biocoenosis” as it is uncommon

Page 4 - Paragraph commencing with “However, certain colonies”  Unclear if referring to colonies in 2019 surveys or earlier surveys? Clarify

Page 6 – How was the orthomosaic created? If using metashape what were the settings used to create the orthomosaic and how was the mosaic scale calibrated? This may be explained in Sandin 2019 but even so, would be helpful to put details here.

Page 7 – unclear how species turnover rates were examined when the three main genera of corals were classified to genus level ?

Page 7 – replace the word “Notorious”

Page 12 – Unclear which appendix 1 referring too?

Results

Page 14 – Not species groups, rather taxa groups. Likely many of the species were missed given the need to groups three genera at genus level

Page 15 – Better to refer to the grouping as “Tax” rather than species

Page 15 section 3.2 – Multivariate analysis of community changes i.e. PCA or MDS plots with permanova tests would be  a better way to demonstrate community level changes.

Page 16 – unclear how it was determined that “The majority of appearing and disappearing species belong to Acroporidae…..” when this taxa was pooled to genera level?

Page 16 – If the most recent taxonomic guide being used is Veron 2000 then the following claim “Taxonomic revisions were taken into ac-count, and species were named according to current taxonomic updates” is not valid as there have been many taxonomic changes since Veron 2000. Perhaps state “ and species were names according to Veron 2000”

Page 17 – Net reef growth over time is arguably the most interesting result. Consider ploacing first in the results and providing more detail on the method used to calculate reef height.

Discussion

Page 17 – first paragraph should contain most interesting results at start of paragraph

Page 18 – In my opinion this is the most interesting result (net reef growth) and should be discussed further

Page 19 – Disturbances. How has the disturbance regime contributed to the changes observed in this study? Unclear how these changes

Page 20 – Conclusions should focus on the science of this study. Best reserve credits for the acknowledgements.

Discussion in general needs re-writing to highlight the important results and how these relate with the environmental changes discussed in the introduction and early discussion.

Figure 4 – consider including the time series of images for each plot, or at least the main plot examined.

Figure 11 – this is a very good figure which needs a more descriptive caption and more discussion in the results. For example, do the colours correspond to the same taxa in the three surveys and where are the colonies which persisted for 39 years located in the figures?

Figure 6 – this is a useful figure for future work. Would be a useful analysis to describe what proportion of colonies were new between earlier surveys and 2019 surveys.

Figure 8 – Nice figure

Comments on the Quality of English Language

The manuscript would benefit from further editing to improve the english

Reviewer 2 Report

Comments and Suggestions for Authors

Dear authors,

I carefully read your manuscript and attached here the file with my comments and suggestions. There are very few but, in my opinion, needed revisions to improve the final message of your work and results and some corrections to be more rigorous in the use of some terms.

I really appreciated this work and the great opportunity you had to re-monitor a quite pristine coral-reef site. It is exactly in this aspect that lies the great potential of the manuscript. I think that you should go deeper in this in the Discussion (see the file attached).

Anyway, I think that the manuscript can be consider after minor revisions.

Kind regards

Round 2

Reviewer 1 Report

Comments and Suggestions for Authors

The authors have done an excellent job in reviewing their paper, it was  a pleasure to read. I have no further recommendations other than perhaps  placing some methods and results into a supplementary file to reduce the length of the paper.

Author Response

The authors are grateful for the positive feedback from the reviewer. We appreciate your acknowledgment of our efforts in crafting a comprehensive paper. Your suggestion to relocate certain methods and results to a supplementary file to streamline the length of the paper is duly noted and will be taken into consideration. Thank you for your valuable input and constructive criticism.